# ICMR's multistate implementation research study on integration of screening and management of mental and substance use disorders with other non-communicable diseases (ICMR-MINDS) – An implementation research study protocol

Johnson-Pradeep Ruben [1]‡, Yatan Pal Singh Balhara [2]‡, Ajay Chauhan [3], Abhishek Ghosh [4], Limalemla Jamir [5], Anindo Majumdar [6], Arpit Parmar [7], Debasish Basu [4], Debadatta Mohapatra [7], Prem Mony [8], Chirag Parmar [3], Sharad Philip [9], Renjith-R. Pillai [4], Shankar Prinja [10], Neha Purohit [10], Siddharth Sarkar [2], Roshan Fakirchand Sutar [11], Pulkit Verma [12], Neha Dahiya [13]*, Ashoo Grover [13], on behalf of ICMR MINDS Study Group,¶

1 Department of Psychiatry, St. John's Medical College, Bengaluru, India, 2 National Drug Dependence Treatment Center, All India Institute of Medical Sciences, New Delhi, India, 3 Gujarat Institute of Mental Health, Hospital for Mental Health, Ahmedabad, Gujarat, India, 4 Department of Psychiatry, Addiction Psychiatry and Psychiatry, Postgraduate Institute of Medical Education and Research, Chandigarh, India, 5 Department of Community and Family Medicine, All India Institute of Medical Sciences (AIIMS), Guwahati, Assam, India, 6 Department of Community & Family Medicine, All India Institute of Medical Sciences, Bhopal, India, 7 Department of Psychiatry, All India Institute of Medical Sciences, Bhubaneswar, India, 8 Division of Epidemiology, Biostatistics and Population Health, St John's Research Institute, Bengaluru, Karnataka, India, 9 Department of Psychiatry, All India Institute of Medical Sciences (AIIMS), Guwahati, Assam, India, 10 Department of Community Medicine & School of Public Health, Post Graduate Institute of Medical Education and Research (PGIMER), Chandigarh, India, 11 Department of Psychiatry, All India Institute of Medical Sciences, Bhopal, India, 12 Informatics and Data Centre, Indian Council Medical Research (ICMR), Headquarters, New Delhi, India, 13 Division of Delivery/Implementation Research, Indian Council of Medical Research (ICMR), Headquarters, New Delhi, India

¶Membership of the **ICMR MINDS Group** is provided in the Acknowledgements.
☙ These authors contributed equally to this work.
‡ JPR & YPSB joint lead authorship on this work.
* drnehadahiya@gmail.com

## Abstract

Non-Communicable Diseases (NCDs) are now a leading cause of mortality and morbidity globally, and mental illness is a significant part of it. In India, the treatment gap for common mental disorders is over 80%. In order to bridge this gap, mental health treatment models recommend task-shifting to non-specialists and integration of mental health care into general healthcare services. Other NCDs are being managed effectively by non-specialist healthcare workers (HCWs) at primary care, and mental illness and substance misuse are highly comorbid with other NCDs; hence, integrating mental health care within the NCD services and care framework seems logically feasible and effective. However, country-specific characteristics pose a significant challenge to the implementation of integrated care for mental disorders and NCDs. The primary objective of this study includes the development and implementation

**Data availability statement:** Deidentified research data will be made publicly available when the study is completed and published. No datasets were generated or analyzed during the current study phase. All relevant data from this study will be made available upon study completion.

**Funding:** This study is funded by the Indian Council for Medical Research (ICMR), Headquarters, New Delhi. The funder was involved in identifying the research priority, developing the protocol, and reviewing the manuscript. However, the funder will not be involved in data collection, analysis and interpretation of the results.

**Competing interests:** We declare no competing interests. N.D., A.G., and P.V. are staff members of ICMR. The opinions expressed by them in this paper are their own and do not necessarily reflect the policy of ICMR. The study coordinator, Dr Neha Dahiya, Scientist, Delivery Division, ICMR Hqrs, will, however, oversee that the timelines are met and the study is being conducted as per protocol and following the ethical standards. This does not alter authors' adherence to PLOS ONE policies on sharing data and materials.

of a service delivery model that would result in at least 70% coverage of screening, linkage to care, and management of common mental disorders and substance use disorders (MSUD) among persons seeking care for NCDs at public health facilities. Secondary objectives include assessment of the feasibility of adoption of the implementation model by the health care system and to evaluate the cost of the mental health service strengthening intervention package from the health system's and the patient's perspectives. It will be a multi-site implementation research study, employing a mixed-methods quasi-experimental, within-site, three-phase, single-arm, interrupted time series design. The implementation model comprises screening, treatment, and linkage of mental health services integrated into NCD care in at least three blocks in each of the seven selected districts of the seven selected states of India, which are geographically far apart. The expected outcome would be to increase the proportion of patients screened and managed for MSUDs among persons seeking care for NCDs at the public health facilities. The results of this implementation research will provide a roadmap for scaling up of integrated MSUDs services within general healthcare.

## Trial registration

ClinicalTrials.gov CTRI/2024/08/072748.

## Introduction

Non-communicable diseases (NCDs) are the leading causes of morbidity and mortality globally, accounting for nearly two-thirds of deaths. In India, NCDs account for 16,939 disability-adjusted life years (DALYs) per 100,000 population, alongside high out-of-pocket and catastrophic health expenditures [1]. The co-occurrences of NCDs and, mental and substance use disorders (MSUDs) significantly impact treatment adherence and health outcomes. Multimorbidity of physical and MSUDs has been previously described as a 'rule' [2].

MSUDs, including depression, anxiety, alcohol use disorders, and tobacco use disorders, contribute substantially to this burden, often co-occurring with chronic conditions like cardiovascular diseases and diabetes [3]. For instance, individuals with alcohol misuse have a 1.4 times higher likelihood of developing heart disease, while anxiety disorders increase this risk to 2.2 times [4]. Moreover, persons with MSUDs are at a higher risk of living with the risk factors for NCDs, including an unhealthy diet and physical inactivity. Common biological mechanisms like immune and inflammatory pathways to shared social determinants (e.g., poverty) could explain the co-occurrence of mental disorders and other NCDs [2].

Studies indicate that the co-existence of NCDs, such as hypertension and diabetes, with depression and other mental disorders is detrimental to care and prognosis, leading to poor glycaemic control, uncontrolled hypertension, greater risk of

cardiovascular complications, and higher mortality rates [5]. As per the United Nations Development Programme, of the estimated 15 million excess deaths over 2020–2021, a significant portion were likely to be people living with NCDs. A survey conducted by the World Health Organization (WHO) across 60 countries found that 9.3–23% of patients with chronic diseases had comorbid depression [6]. Moreover, NCDs affect one's mental well-being and functioning, impacting treatment adherence, and MSUDs are risk factors for NCDs. There is a need for immediate, focused attention to both MSUDs and NCDs [7,8]. Scaled-up investment for mental health and other NCDs is crucial. In addition, integration of care for MSUDs, in NCD care is the need of the hour.

In 2017, NCDs constituted 46.6% of national DALYs, with neuropsychiatric conditions among the top five contributors to disability [9]. Studies indicate that comorbidities such as hypertension and diabetes, when coupled with depression, worsen prognosis and increase mortality risks [10,11]. However, MSUDs services in the country are hindered by systemic barriers at various levels, including service provider factors [12], cultural factors, and sociopolitical factors [13].

Mental health programmes and activities in the country have been described as fragmented and disorganized and have a low priority during implementation, as they suffer from administrative, technical, and resource constraints and, thus, result in limited reach [14]. The poor infrastructure and shortage of human resources of the health care system for the treatment of mental disorders have also been documented previously [15]. Despite initiatives like the National Programme for Prevention & Control of Non-Communicable Diseases (NP-NCD) in India, the integration of mental health into NCD care remains inadequate due to systemic barriers such as cultural stigma, geographic constraints, and insufficient infrastructure and trained human resources [16,17]. Currently, India has only two mental health workers and 0.75 psychiatrists per 100,000 population, which is below global average [18]. The treatment gap for mental disorders ranges from 70% to 92%, underscoring the urgent need for integrated healthcare approaches that effectively address both NCDs and mental health [19]. To address these treatment gaps in mental health care in India, a critical research question arises: How can mental health care be effectively integrated into the existing healthcare framework?

While the moment is opportune to integrate mental health care delivery into the NCD care delivery at various levels of health care delivery, the health system needs to be prepared for the proposed integration. The lessons from the NP-NCD and the National Mental Health Programme of India offer valuable resources to plan and implement the proposed integration.

Comprehensive Primary Health Care (CPHC) aimed to provide health services to communities, and the Ayushman Bharat (AB) program was launched to strengthen CPHC services. One of the key components of the AB program is to enable the expansion of the package of services that go beyond maternal and child health to include care for non-communicable diseases, palliative and rehabilitative care, oral, eye, and ENT care, mental health, and first-level care for emergencies and trauma.

While there has been a growing focus on the common behavioural risk factors, including tobacco use, alcohol use, and unaddressed psychosocial stress, evidence-to-practice gaps exist at various levels of implementation, including segregated management protocols, low data quality and reporting, weak supply chains, and low treatment adherence. Also, to ensure a successful integration, the mental health component of CPHC needs to focus on simplifying disorder-related concepts into elements that the health workers can easily and quickly understand; integration of the mental health component into the practicing style of the health workers so that they can pick up mental health issues during their routine practice; task-shifting; using affordable and straightforward technology; and facilitation of promotion of universal prevention-focused interventions common to both mental disorders and other NCDs [20].

Hence, there is a need to plan the integration meticulously to reach the desired goal. This requires structured implementation of the proposed integration using the building blocks of a health system comprising of service delivery, health workforce, health information system, medical products, financing, leadership, and governance [21]. Such a framework would address the gaps in all the tiers of a health system, starting from a sub-center to a tertiary referral health facility. Moreover, the process of proposed integration needs to be studied systematically.

## Aims & objectives

The aims of the study are to develop, assess the feasibility, implement a service delivery model to achieve high coverage (≥70%) of screening, management, and linkage to care of common mental disorders (CMDs) and substance use disorders (SUDs) among individuals seeking care for non-communicable diseases (NCDs) at public health facilities and to evaluate the cost associated with the implementation of the proposed service delivery model.

The primary objective is to develop and implement a service delivery model that would result in high coverage (at least 70%) of screening, management, and linkage to care of common MSUDs among persons seeking care for NCDs at public health facilities. The secondary objectives are to (a) assess the feasibility of adoption of the implementation model by the health care system and (b) evaluate the costs of the mental health service strengthening intervention package from the health system as well as the patient's perspective.

## Materials and methods

### Study design

It is a multi-site implementation research study employing a mixed-methods, single-arm, quasi-experimental interrupted time series design with a pre-post approach conducted within-site over three phases and without a control group. The study would incorporate formative and evaluative components into iterative improvement cycles.

### Study setting

The study will be carried out in seven states across India. The states are self-governing administrative divisions, and each of them has a state government. The governing powers of the states are shared among the state government and the union government. The study in each state shall be overseen by investigators belonging to a local medical institution having a medical school and an attached tertiary care hospital. These states and the institutions offering the oversight include Assam [All India Institute of Medical Sciences (AIIMS), Guwahati], Madhya Pradesh (AIIMS, Bhopal), Gujarat (Gujarat Institute of Mental Health, Ahmedabad, Gujarat), Haryana (AIIMS, New Delhi), Karnataka (St. John's Medical College Hospital, Bengaluru), Odisha (AIIMS, Bhubaneswar), and Punjab [Post Graduate Institute of Medical Education and Research (PGIMER), Chandigarh].

Within each state, one district has been selected for the study. These include Kamrup (Assam), Vidisha (Madhya Pradesh), Gandhinagar (Gujarat), Faridabad (Haryana), Bengaluru-Rural (Karnataka), Khordha (Odisha), and Mohali (Punjab). These districts have been identified in each of the states in consultation with the respective state health department. Intervention will be implemented at all the health facilities in the district. However, evaluation will be done at 81 health facilities in each district. In total, 567 health facilities will be evaluated to assess the implementation outcome, patient outcome, and service outcome. The public health care delivery systems at all levels of care delivery, including Ayushman Arogya Mandirs, sub-centers, Primary Health Centers (PHCs), Community Health Centers (CHCs), District Hospitals, and tertiary care hospitals in rural and urban areas (depending on the need and feasibility), shall be included in the study. The coordinating site would be the Indian Council for Medical Research (ICMR), Headquarters (HQ), New Delhi. The estimated time of completion of data collection will be October 2026, and the results are expected in January 2027.

### Study population

The target population includes individuals aged 18 years and above who seek care for NCDs. The study population shall be the healthcare workers of all cadres across healthcare facilities at various levels of healthcare delivery (including Ayushman Arogya Mandirs, sub-centers, PHCs, CHCs, district hospitals, and tertiary care hospitals)

### Eligibility criteria

**Inclusion criteria.** The healthcare workers aged 18 years and above, of all cadres across healthcare facilities involved with NCD and Mental Health at various levels of healthcare delivery (including sub-centers, AAMs, PHCs, CHCs, district hospitals, and tertiary care hospitals) in the selected districts across the seven states, interested in participating in the study and willing to provide consent for the same, shall be eligible for inclusion in the study.

**Exclusion criteria.** Those refusing to consent or not interested in participating in the study.

### Sample size

One district from each of the seven states shall be included in the study. In each district, blocks (rural areas) and wards (urban areas) shall be listed. A listing of healthcare facilities across each of the districts shall be carried out.

The study shall be carried out in three phases: Phase I (Formative research), Phase II (Model Optimization [Mx]), and Phase III (Full implementation and evaluation).

For **Phase I**, one block or ward from each of the districts shall be included. All the health facilities (including AAMs, sub-centers, PHCs, CHCs, district hospitals, and tertiary care hospitals) in the selected block or ward shall be included. Each block and ward are expected to have one CHC and three to six PHCs. Each PHC is expected to have 5–6 AAMs. The district hospital in each of the districts shall also be included in this phase of the study. A service delivery Model 0 will be developed in collaboration with the key stakeholders, co-development of the implementation model, and the co-developed Model 0 will then be implemented in one block.

In **Phase II**, the implementation model will be refined in the block or the ward based on the challenges and feedback from all the stakeholders. The study will follow an iterative process until the optimized version of the Model Mx is achieved.

In **Phase III**, the final model shall be implemented across the district at each of the sites. The primary evaluation outcome is facility-level coverage of the mental health screening and management strategy. We require an estimate of coverage of 70% with an absolute precision of ±10% and 95% confidence. Using the single-proportion formula $n = Z^2 p (1-p)/d^2$, with $Z = 1.96$, $p = 0.70$, and $d = 0.10$, gives $n = 80.7$, rounded up to 81 health facilities per district. The implementation cascade was designed such that screening, referral, and management rates of ≈90% at each step yield an overall coverage of $0.9 \times 0.9 \times 0.9 = 0.729$ (≈72.9%), which is slightly conservative with respect to the 70% target. Because the primary unit of analysis is the facility, this sample-size calculation does not require an intra-cluster correlation correction; however, if analyses are to be performed at the individual level or if clustering is a concern, the sample will be inflated by an appropriate design effect (DEFF = 1+ (m − 1) ρ), and allowance will be made for anticipated non-response. A sensitivity analysis using DEFF values between 1.2 and 1.5 and a 10% non-response adjustment is presented in Supporting Information S1 Table. (S1 Table – Sensitivity Table using Appropriate Design Effect)

### Sampling method

We will be using a multistage sampling approach that will be adopted to ensure representation across multiple stakeholder groups and healthcare delivery levels. Initially, the seven states were selected by the expression of interest, and later districts were selected after the discussions with the respective states.

In **Phase 1**, purposive sampling will be used to recruit key informants, including policymakers, administrators, healthcare workers, community representatives, and patients, to capture diverse perspectives on mental health integration.

In **Phase 2** and **Phase 3**, all healthcare facilities—Ayushman Arogya Mandirs (AAMs), Sub-centres (SCs), Primary Health Centres (PHCs), Community Health Centres (CHCs), and District Hospitals (DH)—as well as all health cadres involved in mental health service delivery under the District Mental Health Programme (DMHP), including Medical Officers (MOs), Community Health Officers (CHOs), and Accredited Social Health Activists (ASHAs), will be included.

This staged approach allows for systematic recruitment from broader administrative and policy levels down to frontline providers and end-users, facilitating comprehensive data collection across the health system. A flowchart of the multistage sampling method is presented in Fig 1.

### Intervention components

The intervention components of the research study include tools for integrated screening and assessment for mental health conditions; treatment guidelines, workflows, and pathways that integrate mental health interventions within NCD care; the development of a digital platform/app; training for healthcare providers on integrated care practices; and patient education and engagement to raise awareness and promote shared decision-making.

(1) Tools for integrated screening and assessment for mental health conditions shall be developed for all levels of health-care personnel. These shall include the screening tools to be used by Accredited Social Health Activists (ASHA) and Community Health Officers (CHOs) and assessment tools to be used by the Medical Officers (MO). The screening tools shall be developed by the ICMR HQ in consultation with mental health experts. The assessment tools for the MOs shall be developed by the participating sites. These screening and assessment tools shall be developed for emergency mental health conditions, common mental disorders (depressive disorder and anxiety disorder), substance use disorder (tobacco and alcohol use disorder), and self-harm. The proposed screening tool shall be based on the mhGAP intervention guide (by the World Health Organization), Patient Health Questionnaire-9 (PHQ-9) [22], and the

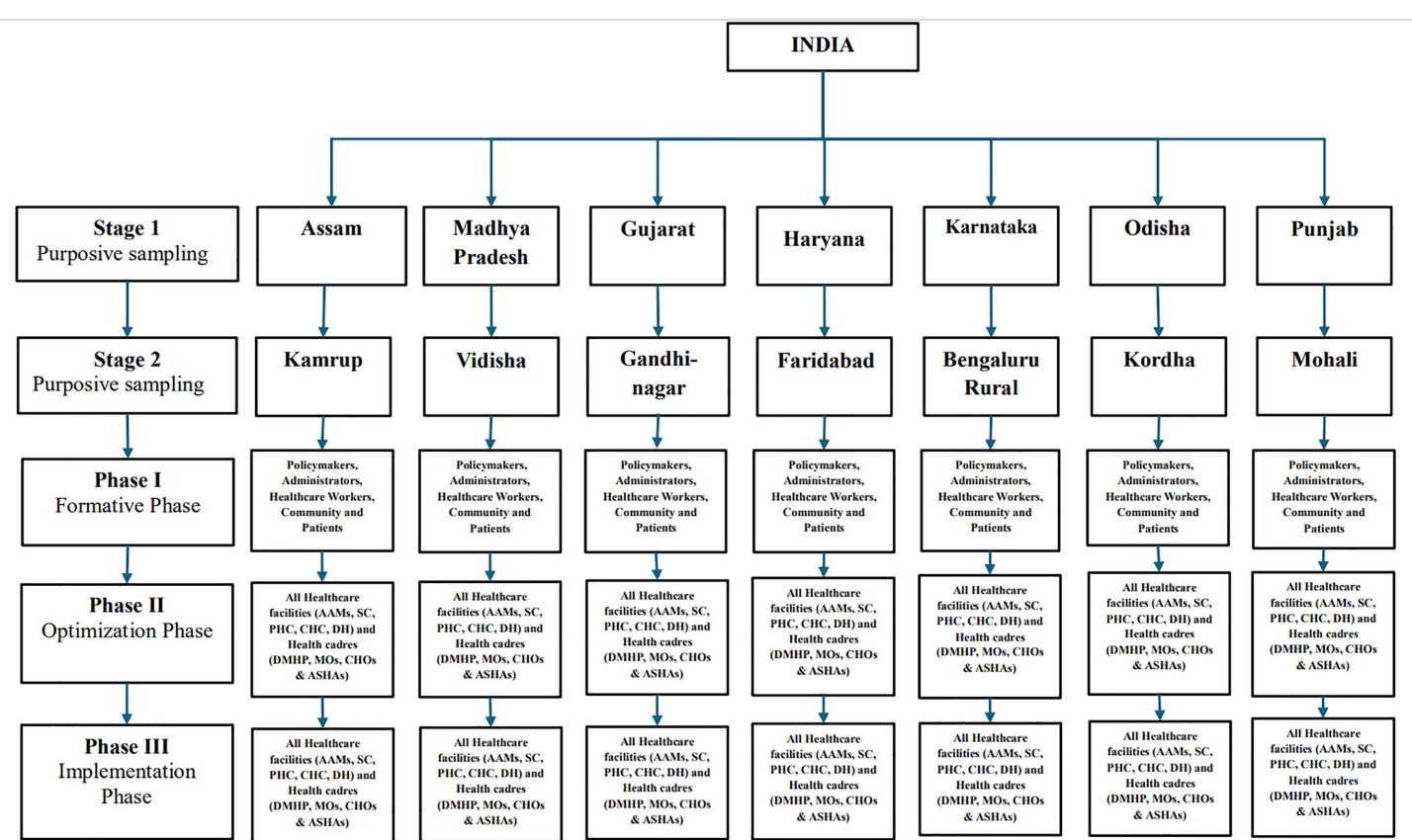

**Fig 1. Flow chart of multiphase sampling of ICMR MINDS.**

Generalized Anxiety Disorder-7 (GAD-7) [23] and shall include a screener for tobacco, alcohol, and other psychoactive substances [24], which will be translated to local languages in each state and validated before using it. The PHQ-9 and GAD-7 are validated scales and available in local languages. The assessment tools shall be based on the recent version of the diagnostic system, viz., International Classification of Diseases (ICD)-11. All the tools will be administered in an interview format.

(2) The workflows and pathways that integrate mental health interventions within NCD care shall be developed. These shall be developed by the ICMR HQ and the study sites. These shall be based on the WHO mhGAP intervention guide [25] and other guidelines on the management of MSUDs. In order to ensure the applicability to the local context, the recommendations made by various organizations, associations, and departments in the county shall be taken into consideration. The ICMR Standard Treatment Workflow of India for Psychiatry [26] and the ICMR Community Health Officers' Mental Health Workflow shall be some of the resources (S1 Fig). Drug interactions shall be considered while drawing up a management plan. Screening, assessment (diagnosis and severity of the disorder), and management (treatment and referral) shall constitute the intervention package. All aspects of the intervention package shall be translated into the local languages.

(3) A digital platform and smartphone app shall be developed. These shall be developed for various purposes, including the digitization of the screening tools, referral and management protocols using an electronic decision support system; tracking of the number of participants screened, treatment initiated in screen positives, number of follow-ups by the participants, number of participants linked to CHC, District, and Hospitals; care coordination mechanisms for effective communication and referral systems using recall and reminder systems; and monitoring and evaluation of the project.

A mHealth App Usability Questionnaire (MAUQ) will be used for assessing the usability of these applications. The MAUQ evaluates mHealth app usability using a set of targeted questions across three main categories, including Ease of Use and Learnability, Interface and User Satisfaction, and Usefulness. The provider version of the questionnaire shall be used for the purpose of the study. The correlation coefficients among the MAUQ, Post-Study System Usability Questionnaire (PSSUQ), and System Usability Scale (SUS) were found to be high [27].

(4) Trainings shall be carried out for the healthcare providers on integrated care practices. Health care professionals across different cadres (including ASHAs, CHOs, Nurses, and MOs) will receive training. The trainings shall be focused on screening and assessment tools, treatment and referral flow, and use of digital platforms/apps. The training shall include the induction training in the beginning, followed by mentoring and ongoing supportive supervision. Cascade trainings shall be carried out based on the modules developed as part of the project.

(5) Content aimed at patient education and engagement to raise awareness and promote shared decision-making shall also be developed and disseminated to the community through an awareness programme.

The key intervention components that shall be delivered at different levels of the health system are provided in Table 1. The health care providers across these settings shall include ANM, CHO, MO, nurses-PHC, and CHC doctors and nurses.

## Status and timeline of the study

The study has been sanctioned for the duration of three years, from 01.02.2024 to 31.01.2027.

**A) Recruitment and Data Collection**

• Recruitment Initiation: Participant recruitment commenced in May 2025 and is ongoing.

• Participant Enrollment Completion: The target number of participants is expected to be enrolled by October 2026.

**Table 1. Key intervention components that shall be delivered at different levels of the health system.**

|  | Subcenter/AAM | PHC | CHC | DH/MC |
|---|---|---|---|---|
| Screening | + | + | + | – |
| Diagnosis | – | + | + | + |
| Treatment | – | + | + | + |
| Linkage | + | + | + | – |
| Training of health care providers | + | + | + | + |
| Use of Digital platform/ App | + | + | + | + |
| Patient education and engagement | + | + | + | + |

Abbreviations: AAM-Ayushman Arogya Mandir; PHC-Primary Health Center; CHC-Community Health Center; DH-District Hospital; MC-Medical College.

**B) Data Collection Completion:** Data collection is anticipated to conclude by October 2026.

**C) Results Availability:** Preliminary results are projected to be available by February 2026. The final results will be ready after two years, i.e., January 2027.

### Phases of Study Implementation

This research will comprise three phases of implementation: Phase 1—Formative Research, Phase 2—Model Optimization, and Phase 3—Full Implementation and Evaluation. The spirit figure of the phases of ICMR MINDS is provided in Fig 2.

**Phase-1-Formative Research.** The formative research phase aims to assess the service availability and health system readiness in terms of infrastructure. It will attempt to identify key facilitators and barriers to integrating mental and substance use disorder (MSUD) care into the broader non-communicable diseases (NCD) framework. This phase involves a comprehensive situational analysis, including desk reviews, SWOT analysis, and stakeholder interviews, which will inform the development and optimization of the intervention model [28–31]. The details of the overview of the situational analysis are presented in Fig 3.

1. *Desk Review*: Desk review will be used for identification of potential facilitators and barriers to integration of care of mental disorders and substance use disorders into care for other NCDs. Multiple resources from Government and other websites will be explored to gather information regarding the same. Information on the domains of prevalence, mortality, morbidity, risk factors, economic burden, care gap, digital intervention, knowledge, attitude & practices related to common NCDs and mental health conditions, governance, human, physical, and intellectual resources, health financing and information, health insurance, and effectiveness of m-health and telemedicine for access to NCD care and mental health care will be collected from various resources.

2. *SWOT Analysis and Pathways to Care Mapping*: Using insights from the desk review and state-level mental health and NCD indicators, we will perform a SWOT (strengths, weaknesses, opportunities, and threats) analysis. This analysis will help identify site-specific advantages and limitations concerning the integration of MSUD and NCD care. Pathways to care mapping will complement the SWOT analysis by visually representing the current care journey for patients with MSUD and NCDs within each healthcare facility. This mapping exercise will highlight potential bottlenecks, missed opportunities for early intervention, and referral gaps, thereby informing the design of an optimized care pathway within the proposed model.

3. *Stakeholder Interviews:* To gain a comprehensive perspective, stakeholder interviews will be conducted through in-depth interviews (IDIs) and focus group discussions (FGDs) with various participants, including patients, healthcare workers, administrators, and policymakers. The IDIs and FGDs will explore stakeholders' perceptions of the current system, their expectations for integrated MSUD and NCD services, and perceived challenges and benefits of the new model.

| | STUDY PERIOD | | |
|---|---|---|---|
| **Phases of the implementation study** | **Phase 1**<br>**Formative Research** | **Phase 2**<br>**Model Optimization** | **Phase 3**<br>**Full Implementation and Evaluation.** |
| **Aim** | Situational Analysis & Development of Model Zero | Iterative process of Model optimization and implementation | Implementation of the final Model |
| **Implementation Taluk/Block** | **Taluk/Block 1** | **Taluk/Block 2** | **Taluk/Block (selected)** |
| **TIMEPOINT**** | 2024 – 2025 | 2025 – 2026 | 2026 – 2027 |
| **ENROLMENT:** | | | |
| **Eligibility screen** | X | X | X |
| **Informed consent** | X | X | X |
| **INTERVENTIONS:** | | | |
| *[Service Delivery Model]* | X | X | X |
| **ASSESSMENTS:** | | | |
| *[Baseline variables of all Health Facilities]* | X | X | X |
| *[Process monitoring]* | X | X | X |
| *[Implementation outcome]* | X | X | X |
| *[patient/clinical outcome measures]* | X | X | X |
| *[service outcomes]* | X | X | X |
| *[costing of intervention package]* | X | X | X |

**Fig 2. Spirit Figure of Phases of ICMR MINDS.**

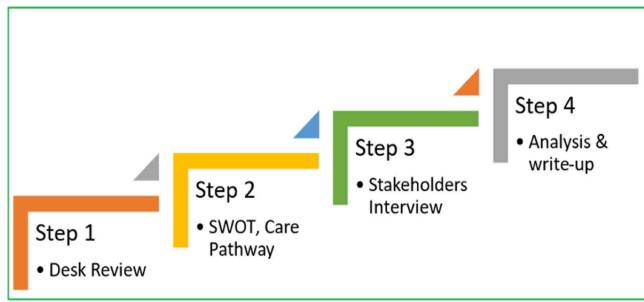

**Fig 3. Overview of Situational Analysis.**

(a) **Focus Group Discussions (FGDs):** A specific guide will be developed for conducting the FGDs of participants. The principle of data saturation will be followed to guide the number of FGDs required at the selected sites. A minimum of four FGDs will be conducted in each of the seven districts. These shall include two FGDs for patients/service users and two for healthcare providers, including CHOs and ASHAs.

**(b) In-Depth Interviews (IDIs):** A specific interview guide will be developed for conducting the IDIs of participants. The principle of data saturation will be used. We will be conducting the IDIs of patients/service users, Health Care Workers (ASHAs, nurses, CHOs, MOs, Psychologists, Psychiatrists and other specialist physicians), health administrators, program managers and policymakers such as Civil surgeon, Chief Medical Officer (CMO), Medical officer In- charge (MOIC) CHCs & PHCs, program manager of National Programme- Non Communicable Diseases (NP-NCD), District Mental Health Program (DMHP)/ National Mental Health Program (NMHP), National Tobacco Control Program (NTCP), District/Block program manager, NGO staff such as that of WHO/NCD Alliance/Voluntary Health Association and local NGOs, members of Alcoholics Anonymous, state-level nodal officers and program officers of NCD, and mental health-related programmes.

4. *Situational analysis write-up:* The initial SWOT analysis and care pathways map collected during the desk-based review and the interviews will form the situational analysis. This phase shall be of particular use for the identification of potential barriers and facilitators for the implementation of the model. We will be using the Consolidated Framework for Implementation Research (CFIR) to identify the determinants of why the baseline rates of screening, assessments, and management are low [32]. The details of the domains and constructs for the ICMR MINDS based on CFIR are provided in Table 2.

1. *Stakeholder Workshop for Co-Development*: Following the situational analysis, co-development workshop will be organized with all key stakeholders, including site investigators, healthcare providers, state health officials, and representatives from ICMR headquarters. During this workshop, findings from the formative research will be discussed in detail, allowing for collaborative decision-making on intervention priorities, care pathways, and service delivery strategies. Through a series of interactive sessions with the stakeholders, Model 0 (M0) will be developed, incorporating their insights to ensure alignment with local contexts and health priorities.

**Table 2. List of the five domains and underlying constructs according to the Consolidated Framework for Implementation Research (CFIR).**

| Intervention Characteristics | Refers to characteristics of the intervention. | • Screening with PHQ-9 (depression), GAD-7 (anxiety), alcohol use screener (alcohol) and tobacco use screener (tobacco)<br>• WHO mhGAP Intervention guide<br>• ICMR standard treatment workflow<br>• Cost of delivering the intervention (time, opportunity cost) |
|---|---|---|
| Individual Characteristics | Refers to characteristics of the individuals who would be key for implementation of an intervention, which include the program managers, health care providers and end users | • Providers<br>• Knowledge and motivation of HCWs for screening and treatment<br>• Training and capacity building of HCWs<br>• Beneficiary<br>• Socio economic status<br>• Family dynamics<br>• Empowerment<br>• Age, sex |
| Outer settings | Refers to the economic, political, social, and cultural context which would influence implementation of the intervention | • Politico-economic context |
| Inner settings | Refers to the health system, and health facility, context which would influence implementation of the intervention. | • Infrastructure<br>• Human resources<br>• Essential medicine supply |
| Process | Focuses on engaging the relevant stakeholders, executing, reflecting, and evaluating with the aim of optimizing the implementation model for the intervention | • Stakeholder Engagement |

The designing, piloting, and refining of the implementation model (M0) for integrating MSUD care into NCD services will be done. This model will be based on findings from the formative research phase, targeting optimal workflows and interventions for screening, assessment, management, and referral of MSUDs within NCD care pathways. The model development will follow an iterative process with continuous feedback loops to optimize the intervention for effective, scalable implementation (Table 1). Key components of this phase include:

**a) Developing Implementation Model 0 (M0):**

The initial implementation model (M0) will be developed as a foundational framework for integrating MSUD services within NCD care. This model will include workflows, protocols, and resource allocation strategies tailored to each level of healthcare delivery, from sub-centers to tertiary hospitals. The model will draw upon existing national guidelines, such as the ICMR workflows and the mhGAP Intervention Guide by the WHO [25], as well as the insights from the formative research regarding local facilitators and barriers. Specific components will include:

- **Screening and Assessment:** Standardized tools based on WHO's mhGAP and ICD-11 for screening and diagnosing MSUDs among patients with NCDs [22–25].

- **Care Pathways:** Well-defined referral pathways and protocols to ensure timely diagnosis, treatment, and follow-up, considering the specific roles of ASHAs, Community Health Officers (CHOs), and Medical Officers (MOs) at different healthcare levels [24].

- **Patient Education:** Materials and engagement strategies will be developed to encourage patient participation in MSUD care.

- **Digital Platform Integration**: A digital application for tracking patients, managing referrals, and monitoring treatment adherence, which will also provide real-time data for evaluation.

**b) Mixed Methods Evaluation Using CFIR and IRLM:**

The implementation model (M0) will be evaluated using a combination of qualitative and quantitative methods, guided by the Consolidated Framework for Implementation Research (CFIR) [32] (Table 2) and the Implementation Research Logic Model (IRLM) (Table 3) [33]. These frameworks will help identify the constructs that contribute to or hinder the integration of MSUD services within NCD care across different settings. The IRLM will support mapping the inputs, activities, outputs, and outcomes, while the CFIR constructs will guide data collection on implementation processes and context.

**c) Logic Model Development for Process Monitoring:**

A logic model will be developed to monitor the implementation processes and outcomes. This model will include:

- Inputs (resources, staff, infrastructure)

- Activities (training, patient screening, referral)

- Outputs (number of patients screened, treated, and followed up)

- Outcomes (improvements in mental health service coverage and patient health metrics). Regular meetings with site investigators and key stakeholders will help refine the logic model based on observed challenges and successes.

**d) Model Optimization through Iterative Pilot Testing:**

The initial model (M0) will be piloted in a sample of healthcare facilities, including selected AAMs, PHCs, CHCs, and Taluk Hospitals (or District Hospitals if available) within the designated districts. This pilot study will allow the research team to

https://doi.org/10.1371/journal.pone.0332359 (PLOS One logo at top)

**Table 3. Pre-facility, facility, and post-facility components of the logic model and specific aspects of the logic model for the implementation framework.**

| | Pre-Facility | Facility | Post-facility |
|---|---|---|---|
| Objectives | Identifying study sites Addressing stigma Human resources | Integrate screening, diagnosis, treatment, and linkage within the health system, monitoring of treatment adherence | Sustainability of screening and treatment for mental health conditions Integrated care being scaled up at the state level |
| Activities | Stakeholder's meetings Community engagement Capacity building workshops | Stakeholder's meetings Involve policymakers Community engagement Capacity building of relevant HCWs, workshops Creating an online platform for seamless data collection and integration | Identify barriers of sustainability and scaling up Collaboration with policymakers and other stakeholders Budgetary allocation |
| Personnel involved | Policymakers, health department, community leaders, HCW | ASHA workers, ANM, CHO, MO Private doctors | Policymakers Health department officials |
| Outcomes | Increased public awareness and reduced stigma Availability of trained human resources | At least 70% screening for target mental health conditions among people seeking care for NCD At least 70% of those screened positive, get appropriate treatment and linkage to care. Screening and treatment are acceptable, feasible, adoptable, and affordable | Adoption and scaling up of the integrated care at the state level |

**Specific aspects of the logic model for the implementation framework**

| Target population | Underlying assumptions | Resources and challenges | Activities (input) | Output | Outcomes |
|---|---|---|---|---|---|
| Persons with NCD seeking treatment at the health facilities Beneficiaries Persons with NCD and comorbid target mental illness and tobacco and alcohol misuse | Theoretical assumption of how the program will work? It is possible to train the existing HCWs and integrate screening, treatment, and linkage to higher services | ANMs, CHO, PHC MOs, GPs (private hospitals) | Training and capacity building for existing HCW cadre to deliver the intervention package Establishing a monitoring system Training of a cadre in the existing healthcare system in monitoring Periodic quantitative and qualitative monitoring Advocacy Additional funds and resource allocation | Number of persons with NCD screened, identified with target mental health conditions, treated, and successfully linked at various health care levels | • % screened • % identified with mental health conditions • %treated as per the guideline • % successfully linked with higher services • % retained in the treatment • Generating a cascade of care with a target at least 70% at each level • Acceptability, feasibility, Adoption, fidelity, implementation cost of the program |

test the feasibility and effectiveness of the intervention package under real-world conditions. (Tables 4 and 5) Key activities during pilot testing will include:

◦ Training and Capacity Building: Health workers (ASHAs, CHOs, and MOs) will receive targeted training on MSUD screening, assessment, and treatment protocols, as well as on the use of digital tools for patient tracking.

◦ Workflow Refinement: Feedback from healthcare providers will be gathered to refine workflows, screening protocols, and referral pathways, ensuring the model is efficient and adaptable across various settings.

◦ Stakeholder Engagement: Regular feedback loops with healthcare providers, administrators, and community representatives will help address practical barriers and enhance local buy-in.

**Phase-2 – model optimization.** In Phase-2, based on the insights gained from pilot testing, subsequent iterations of the model (e.g., M1, M2…Mx) will be developed to improve coverage and quality of MSUD services. Each iteration will incorporate lessons learned from previous cycles, such as adjustments to training content, improvements to digital

**Table 4. Human resources, service delivery, and infrastructure focus during the pre-integration, integration, and post-intervention.**

| | Pre-integration of NCD and mental health care | Integrated NCD and mental healthcare | Post-intervention (sustenance) |
|---|---|---|---|
| **Human resources** | • Training and capacity building of the target human resource<br>• Developing cadres of trainers<br>• Developing a cadre of monitoring masters | • Ongoing training, capacity building, and feedback of newly recruited human resources<br>• Experiential ongoing training by the field research staff | • The master trainer cadre continues the training and capacity-building work. |
| **Service delivery** | | • Screening, treatment, and linkage (STL) to care/adhering to the care cascade for the target mental health conditions in persons with NCD<br>• Integrated and seamless record collection and periodic monitoring (and feedback) | • STL continues<br>• Periodic monitoring continues by the district health department (by the monitoring masters). |
| **Infrastructure/ logistics/ supplies** | • Designing the digital platform/app for seamless and integrated implementation and data collection and monitoring<br>• Developing training modules for master trainers and HCWs | • Human resources (field coordinator for supervision, monitoring, and quality assurance, and for data analysts)<br>• Support to continue to provide incentives to the HCWs<br>• Ongoing technical support for app development, piloting, and maintenance | • Additional budget allocation for sustenance of the service |

**Table 5. Nine aspects of concept mapping as suggested in the Expert Recommendations for Implementing Change (ERIC) that shall be utilized during Phase 2.**

| | |
|---|---|
| **Using evaluative and iterative strategies** | We will be developing a formal implementation blueprint, and it will be re-examined based on the feedback and suggestions given by the stakeholders. We will also be conducting regular audits and adapting the feedback to improve the implementation. We will be developing and organizing quality monitoring systems and implementing tools for the same. |
| **Providing interactive assistance** | We will be providing centralized assistance to all the centers for any technical and implementation aspects. Also, the clinicians will be supported through a collaborative care model. |
| **Adaption and tailoring to context** | We will promote adaptability and tailor implementation strategies based on the expert inputs. |
| **Developing stakeholder interrelationships** | We will be networking with the local panchayat or government leaders and informing the local opinion leaders. We will conduct regular meetings with them and update them about the developments in the implementation. |
| **Training and educating stakeholders** | Workshops/meetings will be conducted involving state government officials and experts from study sites, including regional and district health offices, health facilities, and implementing partners. The workshop will orient stakeholders and strengthen local partnerships for successful implementation of project activities. |
| **Supporting HCWs** | The HCWs will be provided training on the service delivery model, which will include educational materials, treatment guidelines, protocols, communications, digital app and tools for successful implementation. |
| **Engaging patients and caregivers** | Patients and caregivers will be engaged from the formative research till the end of the study to get opinions on the implementation of the service delivery model. Given educational materials and awareness, mental health will be provided. |
| **Utilizing financial strategies** | A review of financial strategies to innovate on financial schemes or incentives to be evaluated in each site shall be considered. |
| **Changing infrastructure** | Necessary changes to the infrastructure in terms of changing physical structure and equipment and recording systems will be requested. |

tracking tools, and enhanced patient engagement strategies. These iterative cycles will continue until a model is identified that demonstrates high coverage, feasibility, and effectiveness across all levels of the healthcare system.

**Continuous stakeholder feedback and model adaptation.** Throughout the optimization process, feedback from all stakeholders, including healthcare providers, patients, and local health officials, will be actively solicited. This feedback will guide adjustments to the model, ensuring it remains responsive to the needs of the healthcare workers and patients. Workshops and collaborative sessions will allow stakeholders to reflect on the model's performance, share experiences, and contribute to its ongoing refinement.

**Integration of CFIR constructs for process evaluation.** During each cycle of model development and optimization, CFIR constructs will guide the process evaluation, ensuring comprehensive assessment of implementation processes. Constructs such as intervention characteristics (e.g., complexity, adaptability), outer setting (e.g., external policies, patient needs), inner setting (e.g., culture, implementation climate), and individual characteristics (e.g., knowledge, beliefs) will help identify factors impacting the effectiveness and sustainability of the model. These evaluations will ensure that any final model is evidence-based, contextually relevant, and scalable across other regions.

**Phase-3-full implementation and evaluation.** The optimized implementation model finalized in Phase II shall be implemented across all the healthcare facilities across all the selected districts. We will continue to use the nine aspects of concept mapping as suggested in the Expert Recommendations for Implementing Change (ERIC) [34] while implementing the optimized model across the selected Taluks or the whole district.

The implementation support team (the research team) will continue to be active in the capacity of facilitator and advisor to the implementation teams (HCWs engaged in care delivery). There will be ongoing mentorship and quality improvement feedback cycles within each facility. Implementation activities will include ensuring access to and use of the intervention package, the digital platform, and the application; HCW training; community awareness and demand generation; establishment of referral and supervision systems; monitoring; and continuous feedback loops.

**Process monitoring.** The program learning team will slowly transition to documenting implementation processes by measuring the process indicators. The Process indicators for service delivery implementation are described in Table 6.

We shall also collect the data with the aim to evaluate the implementation of the optimized model based on the outcome measures. Our evaluation shall be based on the RE-AIM framework (Evaluation) and Implementation Research Logic Model (Evaluation) [33].

## Study outcomes

**Primary outcomes.** The primary outcomes of the study are

- Proportion of individuals screened for CMD and SUD among those seeking care for NCDs

- Proportion of those screened positive who are successfully referred and linked to appropriate mental health or substance use disorder services.

- Proportion of those linked to care who initiate evidence- based treatment and support.

**Secondary outcomes.** The secondary outcomes of the study are

- Proportion of facilities who started integrating the model

- Incremental cost of the implementation of the model

**Outcome measures.** There shall be two sets of outcome measures: key process outcome measures and intervention outcome measures.

Table 6.  Key process indicators for service delivery implementation.

| Component | Purpose | Formative Uses | Summative Uses | Assessment tools |
|---|---|---|---|---|
| Fidelity (quality) | Extent to which intervention was implemented as planned. | To assess the Monitor and adjust program implementation as needed to ensure theoretical integrity and program quality | • Proportion of health facilities integrated with mental health with NCDs | • Quality check using the CHO/MO in each of facility once a month |
| Dose delivered (completeness) | Amount or number of intended units of each intervention or component delivered or provided by interventionists. | To plan number of facilities the service delivery will be implemented | • Number of trainings conducted for ANM/ ASHA on screening and referral for MSUD among patients with other NCDs<br>• Number of trainings conducted for CHO on screening, management and referral for MSUD among patients with other NCDs<br>• Number of trainings conducted for MOs on assessment, management of MSUD among patients with other NCDs<br>• Number of training sessions conducted for ANM/ASHA/ CHO/ community leaders on patient education and engagement to raise awareness and promote shared decision-making | • Numerators and denominators |
| Dose received (exposure) | Extents to which participants actively engage with, interact with, are receptive to, and/or use materials or recommended resources; | To monitor and take corrective action to ensure participants are receiving and/ or using materials/ resources. | • Number of training sessions attended by ANM/ ASHA to do screening and referral for MSUD among patients with other NCDs<br>• Number of training sessions attended by CHO to do screening management and referral for MSUD among patients with other NCDs<br>• Number of training sessions attended by MO to do assessment, management of MSUD among patients with other NCDs<br>• Number of training sessions attended by ANM/ASHA/ CHO/ MO on Digital App<br>• Number of training sessions attended by ANM/ASHA/ CHO/ community leaders on patient education and engagement to raise awareness and promote shared decision-making | |
| Dose received (satisfaction) | Participant (primary and secondary audiences) satisfaction with program, interactions with staff and/or investigators | To obtain regular feedback from health care workers and use feedback as needed for corrective action. | • Pre and post-tests will be conducted using a questionnaire to understand the satisfaction of the Health care workers following the training programme.<br>• Based on the scores obtained by them personalized feedback will be given and booster sessions will be given to those low scores. | • Pre and posttest for ANM/ ASHA training<br>• Pre and posttest for CHO training<br>• Pre and posttest for MO<br>• Pre and posttest for Digital App training for ANM/ ASHA<br>• Pre and posttest for Digital App training for CHO<br>• Pre and posttest for Digital App training for MO |

*(Continued)*

**Table 6.** (Continued)

| Component | Purpose | Formative Uses | Summative Uses | Assessment tools |
|---|---|---|---|---|
| Reach (participation rate) | Proportion of the intended priority audience that participates in the intervention; often measured by attendance; includes documentation of barriers to participation. | Monitor numbers and characteristics of participants; ensure sufficient numbers of target population are being reached. | • Proportion of health facilities from where the HCWs were trained on screening, management and referral for MSUD among patients with other NCDs<br>• Proportion of health facilities from where the HCWs were trained on Digital App<br>• Proportion of ANM/ ASHA trained in screening and referral for MSUD among patients with other NCDs<br>• Proportion of CHO trained in screening, management and referral for MSUD among patients with other NCDs<br>• Proportion of MO trained in assessment, management of MSUD among patients with other NCDs<br>• Proportion of ASHA/ ANM trained on Digital/App<br>• Proportion of CHO trained on Digital App<br>• Proportion of MO trained on Digital App<br>• Proportion of ANM/ ASHA/ CHO community leaders, trained on Patient education and engagement to raise awareness and promote shared decision-making | • Attendance Sheets for all training programs<br>• Reasons documented for why participants did not attend the training |
| Recruitment | Procedures used to approach and attract participants at individual or organizational levels; includes maintenance of participant involvement in intervention and measurement components of study. | Monitor and document recruitment procedures to ensure protocol is followed; adjust as needed to ensure reach. | • Recruitment of the Health care workers for the training will be based on the list provided by the District Health Officer.<br>• Persons aged 18 years and above, being screened for NCDs or diagnosed with NCD | • Number of requests placed with district administration for training of ANM/ ASHA/ CHO/ MO<br>• Proportion of ANM/ ASHA invited to participate in the training sessions<br>• Proportion of CHO invited to participate in the training sessions<br>• Proportion of MO invited to participate in the training sessions<br>• Proportion of ANM/ ASHA contacted telephonically/ personally to ensure participation in the training programs<br>• Proportion of CHO contacted telephonically/ personally to ensure participation in the training programs<br>Proportion of MO contacted telephonically/ personally to ensure participation in the training programs |
| Context | Aspects of the environment that may influence intervention implementation or study outcomes; includes contamination or the extent to which the control group was exposed to the program. | To monitor aspects of the physical, social, and political environment and how they impact implementation and needed corrective action. | • The environmental aspects which can affect the service delivery implementation or outcomes will be ascertained during the formative research and then the solutions would be obtained during the model optimizations and then the intervention will be delivered<br>• Also to prevent contamination, each phase of the study would be done in only one or two blocks/wards | • Number of additional trainings on MSUD carried out in the district besides the training as part of the Project ICMR<br>• Organization that conducted the training<br>• Details of the content, duration, venue (online/ offline), trainers for the training (psychiatrist/ psychologist/ others) disorders covered, type of interventions converted (screening, assessment, management, referral, etc.) for training<br>• Number of the HCWs (ANM, ASHA, CHO, MO) from the block included in the phase II who attended the training |

1. **Key process outcome measures**

The process outcome measures shall include the number of stakeholders sensitized to start integrated mental health services; awareness of HCWs about mental health services integration into NCD services at baseline; number of training sessions conducted; number of HCWs trained for mental health services; acceptance of the training offered to HCWs and feasibility of the interventions; and number of health care workers having the knowledge and motivation to implement the model. (Table 6)

2. **Intervention outcome measures**

The intervention outcome measures shall include implementation outcomes, patient/clinical outcome measures, and service outcomes, which are listed in Tables 7–9.

The process of implementation will be evaluated by the COM-B model of the Behavior Change Wheel (BCW) [35]. As per this model, there are 3 key factors & multiple sub-factors capable of changing behavior, namely capability, opportunity, and motivation.

Program learning shall be documented using the qualitative methods (FGD, IDI), including listing barriers and facilitators, reasons for not following through with treatment, follow-up consultations (provider plus beneficiary), feasibility and acceptability of the training module/program through questionnaires and FGDs, technical and logistical challenges in delivering the program, challenges faced by staff during the completion of the course and their subsequent engagement (reaching out and referring), difficulties encountered when participating in the online refresher training, and the overall satisfaction level of staff post-training.

The timeline for all the study-related activities is provided in the Gantt chart (Table 10).

**Cost analysis of the mental health service strengthening model.** The objective of the cost analysis is to assess the incremental costs associated with the intervention package from both the provider's and patient's perspectives. This will involve determining the costs related to the development, implementation, and scale-up of the intervention package, which aims to integrate the screening, detection, and management of selected MSUDs with NCD care.

The cost analysis aims to determine both financial and economic costs of the mental health service strengthening model. The focus of financial cost analysis will be on identifying and costing the additional resources required for the development and implementation of the intervention package. These estimates will offer essential inputs to inform budgetary planning and resource allocation decisions. In contrast, the economic cost analysis will involve identifying and valuing all resources used in the development and implementation of the intervention, regardless of whether they involve direct financial outlays. This includes existing resources such as infrastructure, personnel time, and donated goods or services, thus capturing the full opportunity cost of the intervention [36]. Cost analysis using an economic standpoint is intended to inform any future economic evaluation, such as cost-effectiveness analysis, should such an assessment be undertaken subsequently.

**Conceptual framework for costing of intervention package.** A stepwise approach will be followed to assess the cost of the intervention package in each district from a healthcare perspective. The activities in the intervention package will be identified as either new activities (initiated as part of the intervention package) or old activities (any existing activities likely to be affected in scale due to the intervention). Subsequently, the new activities (such as digital platforms, etc.) will be costed in three phases during the implementation of the refined model in one block (henceforth referred to as the Mref block). The information on resources used and the associated costs will be acquired through a top-down micro-costing methodology in a retrospective as well as prospective manner. The data related to the start-up costs will be obtained at the beginning of the implementation of the refined model. Since the intervention is likely to require time for uptake (3 months), therefore, a similar data collection on resources and costs related to the intervention package will be conducted at the end of the third month of implementation (early implementation costs) and in the 12th month to estimate the late implementation costs. The outputs related to the intervention will also be determined at the three timepoints. This

**Table 7. Implementation outcome measures.**

| Implementation Outcome | Level of analysis | Methods | Measurements |
|---|---|---|---|
| **Acceptability** (perception among stakeholders (e.g., consumers, providers, managers, policymakers) that an intervention is agreeable) | Facility and Individual provider Individual consumer | • Qualitative assessments using ID and FGD with Stakeholders using tool that is already developed for the project | • Planned after implementation of each iteration of the model |
| **Adoption** (intention, initial decision, or action to try to employ a new intervention) | Facility and Individual provider Organization or setting | • Pre-post items to capture adoption of evidence-based practices | • proportion of facilities where HCWs are screening, for MSUD among patients with other NCDs<br>• proportion of ANM and ASHA doing screening and referral and using the Digital App for MSUD among patients with other NCDs<br>• proportion of CHO doing screening, management and referral and using the Digital App for MSUD among patients with other NCDs<br>• proportion of MO doing assessment and management and using the Digital App for MSUD among patients with other NCDs<br>• proportion of ANM and ASHA who have conducted community awareness/ engagement session on MSUD among patients with other NCDs |
| **Feasibility (Yes/No)** (compatible or actual fit) The extent to which an intervention can be carried out in a particular setting or organization | Facility and Individual providers Organization or setting | • Quantitative assessments from the Healthcare facility registers and government databases. | • Proportion of patients screened for MSUD among those screened/confirmed with NCD (N+M)<br>• Proportion of screened positive/confirmed for MSUD managed, are under follow up, complied with referral |
| **Fidelity** The degree to which an intervention was implemented as it was designed in an original protocol, plan, or policy | Facility and Individual provider (adherence, quality of delivery, program component differentiation, exposure to the intervention, and participant responsiveness or involvement) | • Quantitative assessments from the Health care facility registers and government databases. | • Proportion of facilities who started integrating the model<br>• The proportion of patients who initiated the treatment, following up with the health facility<br>• The proportion of the patients in whom the continuum of care was ensured. |
| **Penetration** (integration of a practice within a service setting and its subsystems) | Organization or setting Facility level: The integration of MH screening in NCD within public health system; (the extent to which MH screening, linkages and treatment completed) | • Quantitative assessments from the Health care facility registers and government databases. | • Proportion of NCD patients screened for MSUD<br>◦ numerator assessed with patients registered in the digital App/ record review<br>◦ denominator assessed with HWC/PHC/CHC register (patients with NCD registered at the facility)<br>• Among those with a positive MSUD screen<br>◦ proportion who are screened, managed and referred<br>◦ numerator assessed with patients registered in the digital App/ record review<br>◦ denominator assessed with HWC/PHC/CHC register (patients with NCD registered at the facility) |

*(Continued)*

**Table 7.** (Continued)

| Implementation Outcome | Level of analysis | Methods | Measurements |
|---|---|---|---|
| **Sustainability** (extent to which a newly implemented treatment is maintained or institutionalized within a service setting's ongoing, stable operations) | Administrators Organization or setting (1) passage (a single event such as transition from temporary to permanent funding) (2) cycle or routine (i.e., repetitive reinforcement of the importance of the evidence-based intervention through including it into organizational or community procedures and behaviours, such as the annual budget and evaluation criteria), (3) niche saturation (the extent to which an evidence-based intervention is integrated into all subsystems of an organization) | • Quantitative assessments from the health care facility registers and government databases. | • Proportion of health facilities that continued the service delivery model developed by this study (also note the number of weeks for which the model was used after it was implemented) • Proportion of ANM/ ASHA continuing to screen, and refer for MSUD among patients with other NCDs • Proportion of CHO continuing to screen, manage and refer for MSUD among patients with other NCDs • Proportion of MO continuing to assess, manage and refer for MSUD among patients with other NCDs • Proportion of ANM/ ASHA/ CHO/ MO continuing to use digital App |
| **Impact** (long term effect of the implementation) | Facility and Individual provider Organization or setting | • Quantitative assessments from the health care facility registers and government databases. • Qualitative assessments from Patient/ Service users | • Proportion of facilities offering screening, management and referral services for MSUD among patients with other NCDs • Experience/ satisfaction of HCWs with the delivery of integrated services using IDI at all levels of facilities • Experience/ satisfaction of patients with the integrated services using IDI at all levels of facilities |
| **Implementation Cost** (direct, indirect, training, logistics) | Organization or setting | Quantitative assessments from the cost involved in delivery of the Interventions from all the Health care facilities | Incremental cost of the implementation of the service delivery model |

process will help estimate the total cost of implementing the intervention in its first year, with late implementation costs aiding in the estimation of the ongoing costs to sustain the intervention model.

Additionally, the activities identified as 'old activities' will be costed at two cross-sectional timepoints (baseline and endline). The baseline costs will reflect the cost of activity without implementation of the intervention package. Therefore, this data collection will be conducted in a non-intervention block (future Mref block) during the implementation of the first test model (M0). Information regarding inputs as well as outputs will be acquired along with the related costs in order to determine the per-unit cost of activity (without the intervention package in place). Since these activities are already established, it is assumed that they will experience an immediate scale-up once the Mref model is implemented. Therefore, we plan to collect the data on inputs and outputs only in the last month of the implementation of Mref. The cost difference between the baseline and endline data will reflect the incremental cost associated with increasing the intensity and coverage of services. Combining the costs of new activities with the incremental cost of old activities at the achieved coverage level will enable the calculation of the unit cost for the mental health service strengthening model.

Finally, using the patient's perspective, the out-of-pocket expenditure to access mental health care services (screening/ management) will be determined at a single time-point across different levels of care (Fig 4).

**Table 8. Patient/Clinical Outcome Measures.**

| 1. | Proportion of patients screened by ANM/ASHA for mental disorders and substance use disorders among those seeking care for NCDs |
|---|---|
| 2. | Proportion of screened positive patients by ANM/ASHA for mental disorders and substance use disorders among those seeking care for NCDs that were referred for further evaluation/ management |
| 3. | Proportion of patients screened by CHO for mental disorders and substance use disorders among those seeking care for NCDs |
| 4. | Proportion of screened positive patients by CHO for mental disorders and substance use disorders among those seeking care for NCDs that were offer intervention and/ or referred for further evaluation/ management |
| 5. | Proportion of screened positive patients for mental disorders and substance use disorders among seeking care for NCDs that were assessed by MO |
| 6. | Proportion of screened positive patients for mental disorders and substance use disorders among those seeking care for NCDs that were started on treatment by MO |
| 7. | Proportion of patients with MSUD who were back referred to CHO that followed up with CHO |
| 8. | Proportion of those screened positive for mental disorders and substance use disorders assessed and managed (treated and/or referred) and are under follow-up care at 6 and 12 months compared to the baseline. |
| 9. | Number of teleconsultations through DMHP and Tele MANAS utilized. |
| 10. | Proportion of drop outs (patients who do not reach the recommended next referral point). |
| 11. | Number of patients diagnosed with MSUD among patients with other NCDs |

**Table 9. Service outcome measures.**

| |
|---|
| 1. Proportion of PHC that have at least one medicine to treat depressive disorder |
| 2. Proportion of PHC that have at least one medicine to treat anxiety disorder |
| 3. Proportion of PHC that have at least one medicine to treat alcohol use disorder (short term) |
| 4. Proportion of PHC that have at least one medicine to treat alcohol use disorder (long term) |
| 5. Proportion of PHC that have at least one medicine to treat tobacco use disorder |
| 6. Proportion of CHC that have at least one medicine to treat depressive disorder |
| 7. Proportion of CHC that have at least one medicine to treat anxiety disorder |
| 8. Proportion of CHC that have at least one medicine to treat alcohol use disorder (short term) |
| 9. Proportion of CHC that have at least one medicine to treat alcohol use disorder (long term) |
| 10. Proportion of CHC that have at least one medicine to treat tobacco use disorder |
| 11. Does the district hospital have at least one medicine to treat depressive disorder? |
| 12. Does the district hospital have at least one medicine to treat anxiety disorder? |
| 13. Does the district hospital have at least one medicine to treat alcohol use disorder (short term)? |
| 14. Does the district hospital have at least one medicine to treat alcohol use disorder (long-term)? |
| 15. Does the district hospital have at least one medicine to treat tobacco use disorder? |
| 16. Proportion of PHC that have routine blood tests (LFT, KFT) |
| 17. Proportion of CHC that have routine blood tests (LFT, KFT) |
| 18. Does the district hospital have routine blood tests (LFT, KFT)? |
| 19. Proportion of health facilities that have IEC material on MSUD available and displayed |
| 20. Number of mental health awareness sessions held (at facility/in community) |

**Table 10. Gantt Chart of the ICMR MINDS.**

| Project Activities | | 2024 | | 2025 | | 2025 | | 2026 | | 2026 | | 2027 | |
|---|---|---|---|---|---|---|---|---|---|---|---|---|---|
| | | Q 1 | Q 2 | Q 3 | Q 4 | Q1 | Q2 | Q3 | Q4 | Q1 | Q2 | Q3 | Q4 |
| 1.1 | Ethical clearance | ■ | | | | | | | | | | | |
| 1.2 | Registration of trial with CTRI | ■ | | | | | | | | | | | |
| 2.1 | Formative Phase | | ■ | ■ | ■ | | | | | | | | |
| 2.2 | Implement Initial MD in NCD Integration Model-0 in Block 1. | | ■ | ■ | ■ | ■ | ■ | | | | | | |
| 2.3 | Implement modified MD in NCD Integration Model-1 in Block 2. | | ■ | ■ | ■ | ■ | ■ | | | | | | |
| 2.4 | Implement modified MD in NCD Integration Model-X in Block. | | ■ | ■ | ■ | ■ | ■ | | | | | | |
| 2.5 | Program Learning Activities | | ■ | ■ | ■ | ■ | ■ | | | | | | |
| 3.1 | Implement the best model of MD in NCD Integration of Model-X in 2/3 new blocks. | | | | | | | ■ | ■ | ■ | ■ | | |
| 3.2 | Process-monitoring activities | | | | | | | ■ | ■ | ■ | ■ | | |
| 3.3 | Outcome measurement activities | | | | | | | ■ | ■ | ■ | ■ | | |
| 3.4 | Development of scale-up plans | | | | | | | ■ | ■ | ■ | ■ | | |
| 3.5 | Final analysis | | | | | | | | | | | ■ | ■ |
| 3.6 | Dissemination of results | | | | | | | | | | | | ■ |

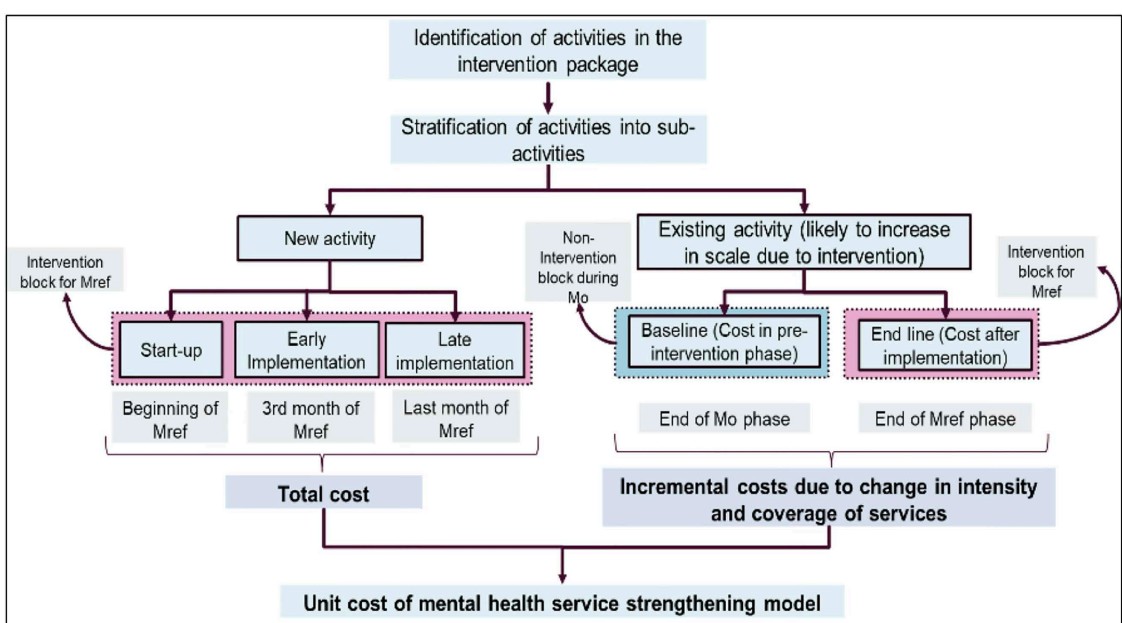

**Fig 4. Conceptual framework for costing of intervention package.**

## Data Collection

A top-down micro-costing method will be employed. Simple random sampling will be used to select primary [2], secondary [1], and tertiary care facilities [1] per district for data collection on the costing of all the components of the intervention package. In total, 28 facilities will be surveyed to gather data on the inputs and outputs associated with the components of the intervention package.

A semi-structured data collection instrument will be developed, tailored to the components of the refined intervention package. This tool will enable the collection of information on the number and types of services provided for the selected

mental health disorders, as well as the value of input resources, including personnel, building space, equipment, furniture, medicines, consumables, and utilities. A structured time allocation tool will also be developed to capture the time spent on various activities by each staff member involved in the delivery of mental health services during the reference year. Data will be gathered through face-to-face interviews with healthcare personnel engaged in different activities, non-participant observations during service delivery, and a review of relevant records.

The details of out-of-pocket expenditure will be obtained through exit interviews at randomly selected primary, secondary, and tertiary care facilities in each district. A total of 200 patients and 20 patients of selected MSUDs visiting outpatient department and inpatient department respectively in the secondary and tertiary care facilities, and 120 individuals visiting the primary healthcare facilities for seeking care for these disorders will be recruited in a consecutive manner. After acquiring an informed written consent, face-to-face interviews will be conducted using a semi-structured instrument, comprising of socio-demographic details, clinical characteristics, and out-of-pocket expenditure. Details on components of both direct medical and non-medical expenditure will be elicited from the participants. Direct medical expenditure will include expenses incurred on the user fee, diagnostic tests, drugs, consumables etc. The direct non-medical expenditure will include expenses on transportation, accommodation, and food for the patient as well as their acquaintances, spent for accessing the mental health services [37].

### Data analysis

The data on the quantity and associated costs of input resources will be recorded in Microsoft Excel. The costs of shared resources, such as human resources, space, equipment, and furniture, will be allocated to specific services using appropriate apportioning methods. The total and per-unit incremental costs incurred, along with their distribution across the various types of inputs and services provided at each facility, will be determined. Additionally, the mean out-of-pocket expenditure per individual will be calculated based on the type of service accessed (screening or management of selected disorders).

### Data governance (Collection, Management, Confidentiality, Security, and Quality Control)

The study will collect both qualitative and quantitative data as a part of a mixed-methods approach to capture primary and facility-level secondary data. The data shall be collected using the direct interviews of health facility-level persons engaged in planning and administration of the health care delivery systems, HCWs, and the persons being screened for or diagnosed with NCDs (quantitative and qualitative) and records available at various healthcare facilities. Data with respect to the screening of MSUDs will occur at multiple levels across the complete continuum of care, including initial screening, diagnosis, treatment, referrals, and follow-ups through a Clinical Decision Support System (CDSS) facilitated through a mobile application. Further, the data related to capacity-building-related activities and outcome measures on the training activities of CDSS will be managed through REDCap [38,39].

Data for implementation outcome measures shall be collected from the health facilities. The details of the same have been provided in Table 8. The data for clinical outcome measures shall be collected from the HCWs and the persons screened for and diagnosed with NCDs. The data for the process outcome measures shall be collected from the health facility level, persons engaged in planning and administration of the health care delivery systems, and HCWs. The nature of data collected from each of the aforementioned shall be based on the outcome measures that have been described in the section on outcome measures.

The mobile application platform will be rolled out in the form of hybrid architecture supported by open-source platforms for both the frontend as well as the backend server-end processing. The frontend of the digital platform will be powered by the cross-platform framework flutter for providing seamless connectivity across various devices. The application backend will be supported by an open-source, high-level Python framework renowned for its scalability, reliability, and rapid development.

In order to accommodate the multiple model refinements inherent in the implementation research project, the software changes will be managed through an agile software development framework with stringent version control for all the software artifacts. The code base changes will be managed through an open-source version control system with the adoption of zero-downtime deployment strategies during the feature rollouts and upgrades. The complete platform will undergo a comprehensive security audit through a CERT-In-empaneled agency for assessing the vulnerabilities at multiple levels. The deployment of the application will be at the Data Centre of the Indian Council of Medical Research (ICMR), Headquarters with the provision of a scalable environment for upscaling the resources on a requirement basis in order to handle varying loads without performance degradation. The data center will facilitate redundant compute and storage components to minimize downtime due to hardware failures and real time data replication along with automated remote backup functionality.

The identity of individual participants will be kept confidential and in adherence to the principle of data minimization, only the bare minimum participant data will be collected through the mobile application, and sensitive data will undergo pseudonymization before being transmitted over the network. The data will be transmitted between mobile application and the server using end-to-end encryption using highly secure protocols like Hypertext Transfer Protocol Secure (HTTPS) and Transport Layer Security (TLS). No data will be stored on the mobile devices, while sensitive data at rest will be stored using 256-bit encryption. The access to data will be strictly through the mobile application controlled through a strict multi-factor authentication (MFA) mechanism along with role-based authorization for ensuring that the users have access to data for which they are authorized.

The access to mobile applications will be monitored through audit logs for any unauthorized access attempts, suspicious actions, or data breaches. All the stakeholders will be trained in capacity-building training for the data security best practices before using the mobile application, and comprehensive data management guidelines will be crafted detailing the procedures for data handling, storage, and accessibility.

The quality control measures that shall be put in place shall include the built-in validation rules, auto-calculation logics, and mandatory field checks, including the study related semantics checks. The respective rules will be appropriately incorporated in the mobile application as well as the REDCap. The data captured through all the sources will undergo regular audits for consistency, missing values, and other study-specific validation rules. Deidentified research data will be made publicly available when the study is completed and published.

## Statistical analysis

**Quantitative analysis.** Data analysis would be done with IBM SPSS software (Version 25.0) and R software (Version 4.5.1). Nominal or categorical variables would be summarized as frequency and percentage. Continuous variables will be summarized as mean and standard deviation when normally distributed and as median and interquartile range (IQR) when non-normally distributed. We shall assess the percentage of the cases seeking NCD care who shall be offered screening, linkage to care, and management of common mental disorders and substance use disorders following the implementation model. Appropriate statistical analyses will be conducted based on the type of data and study design. Comparisons of mean scores between two independent groups will be performed using the independent samples t-test or the nonparametric equivalent if the assumption of normality is violated. For comparisons involving more than two independent groups, one-way ANOVA will be employed. To assess changes in outcome measures over time within subjects, one-way or two-way repeated measures ANOVA will be used, depending on the number of within- and between-subject factors. For categorical variables, associations between independent groups will be examined using the Chi-square test, while the McNemar test will be applied to assess changes in paired categorical data (e.g., pre- and post-intervention). Comparisons of paired quantitative data, such as baseline and post-intervention measurements within the same group, will be conducted using the paired t-test or its nonparametric alternative, such as the Wilcoxon signed-rank test, depending on data distribution. A two-tailed test will be employed. $P < 0.05$ will be taken to be the statistical

significance level. If clustering of variables is detected during analysis, cluster analysis techniques will be employed to explore these patterns.

**Qualitative analysis.** The transcripts from the FGDs and IDIs will be independently translated and transcribed. The results from the IDIs and FGDs would be triangulated. ATLAS.ti (Version 24) will be used for qualitative data analysis, as well as data management.

Both deductive and inductive approaches to qualitative data analysis will be used. Inductive analysis will be done through reading the interview transcripts, and codes and subcodes will be generated. More precisely, thematic content analysis will be used. Further, from these codes and sub-codes, themes and sub-themes will be generated. Verbatim quotes will be mentioned in the final reports under themes and subthemes. In addition, the data collected from interviews with other stakeholders shall be summarized and presented.

**Economic analysis.** Microsoft Excel 2021, along with TreeAge software, is proposed for carrying out the proposed economic evaluation.

## Ethical considerations and declarations

**Approval from the state health departments.** Permission for conducting the study shall be obtained from the respective health departments of each State included in the study.

**Ethics committee approval.** The research protocol has been approved by the 7 institutional ethics committees situated at All India Institute of Medical Sciences (AIIMS), Guwahati; All India Institute of Medical Sciences (AIIMS), Bhopal; All India Institute of Medical sciences (AIIMS), New Delhi; Gujarat Institute of Mental Health, Ahmedabad, Gujarat; St. John's Medical College Hospital, Bengaluru; All India Institute of Medical Sciences (AIIMS), Bhubaneswar; and Post Graduate Institute of Medical Education and Research (PGIMER), Chandigarh. The research will be carried out in alignment with the principles of the Helsinki Declaration and after obtaining clearance from the Institutional Ethics Committee. Important protocol modifications (e.g., changes to eligibility criteria, outcomes, and analyses) to relevant parties (e.g., investigators, the Institutional Ethics Committee, trial registries, journals, and regulators) will be communicated. Written informed consent will be obtained from the participants. Protocol amendments will be submitted to the respective IECs.

## Trial registration

The trial component of the study has been registered with the Clinical Trials Registry of India (CTRI) with the ID No. CTRI/2024/08/072748.

**Informed consent.** The principal investigator and research staff shall obtain written informed consent from all the HCWs and the patients/caregivers prior to their inclusion in the study. We will brief every participant on their prerogative to exit the study whenever they choose. We will share the contact information for mental health services and helplines with the study participants. We will obtain informed consent for sharing the data from each participant. The dataset will be suitably anonymized to prevent any direct association with the identities of the individual participants to protect their privacy.

**Management of adverse events.** Participants are anticipated to face minimal risk, and there's a low probability of untoward incidents. If any participant manifests signs of distress during the study, they will be presented with the choice to terminate their involvement in the program and will be recommended to seek mental health support.

## Discussion

The implementation of integrated care for mental morbidities and other NCDs at the lowest level of healthcare, ensuring at least 70 percent screening, treatment, and linkage to higher levels of care, will ensure holistic care of a person suffering from NCDs and MSUDs and reduce the morbidity and mortality associated with it. At the policy

level, integration of the NMHP and NP-NCD Training and capacity building will lead to generating skilled human resources across healthcare levels. Also, strengthening the linkage between different levels of care in the healthcare system and strengthening the relationship between academia and administration to minimize the evidence-to-practice gap for sustainability, scalability, and monitoring. Therefore, a cohesive and integrated health systems management is essential for the care of co-occurring mental disorders and NCDs that cause substantial, preventable morbidity and mortality. This requires a structured implementation of the levels of prevention (primordial, primary, secondary, and tertiary) on the six building blocks of a health system (World Health Organization) comprising service delivery, health workforce, health information system, medical products, financing, leadership, and governance. Such a framework would address the gaps in all the tiers of a health system, starting from a sub-center to a tertiary referral health facility.

The current study has certain limitations, which include implementation research limited to certain states and blocks in a district; the NCDs will be restricted to Diabetes Mellitus, Cardiovascular diseases including Hypertension, Stroke, Cancers, Chronic Kidney Disease, and Chronic Obstructive Pulmonary Disease, the MSUDs are restricted to Depression, Anxiety, Tobacco use disorders and Alcohol use disorders; and the research shall focus predominantly on the government-run health facilities.

The dissemination plans include publishing the results after appropriate approvals, including professional, public, and private avenues such as academic journals, seminars, conferences, and workshops. After the data collection, we intend to publish multiple articles based on the different phases of the study.

## Conclusion

The existing protocol is a multi-site intervention comprising various service delivery components that employ implementation research methodologies to integrate mental health into other non-communicable diseases within the public health system in selected states of India. The objective is to enhance the coverage of at least 70% for screening, linkage to care, and management of common mental disorders and substance use disorders (MSUD) among individuals seeking treatment for non-communicable diseases (NCDs) at public health institutions. The study will evaluate the feasibility of adopting the implementation model within the public health care system and estimate the costs of the mental health service enhancement intervention package from both the health system and patient perspectives.

## Supporting information

**S1 Fig. ICMR Community Health Officers' Mental Health Workflow.**
(PDF)

**S1 File. Abbreviations.**
(PDF)

**S1 Table. Sensitivity Table using Appropriate Design Effect.**
(PDF)

**S1 Checklist. SPIRIT Checklist for the ICMR MINDS.**
(PDF)

## Acknowledgments

The team is grateful to the experts from across India for their invaluable support. We also extend our heartfelt thanks to the study mentor group, all the study participants, and the Governments of Assam, Gujarat, Haryana, Karnataka, Madhya Pradesh, Odisha, and Punjab for their cooperation and active involvement.

## ^ ICMR MINDS Group – (Alphabetical order)

**Authors: Abhishek Ghosh** – Additional Professor, Postgraduate Institute of Medical Education and Research, Chandigarh. **Ajay Chauhan**– Superintendent, Gujarat Institute of Mental Health, Ahmedabad, Gujarat. **Anindo Majumdar –** Additional Professor, All India Institute of Medical Sciences (AIIMS), Bhopal. **Arpit Parmar** – Associate Professor, All India Institute of Medical Sciences (AIIMS), Bhubaneswar. **Ashoo Grover** – Head, Division of Delivery Research, Indian Council of Medical Research, New Delhi. **Bhupen Barman**– Professor, All India Institute of Medical Sciences (AIIMS), Guwahati. **Biswa Ranjan Mishra** – Professor, All India Institute of Medical Sciences (AIIMS), Bhubaneswar. **Chirag Parmar**– Consultant Psychiatrist, Gujarat Institute of Mental Health, Ahmedabad, Gujarat, India. **Debadatta Mohapatra** – Associate professor, All India Institute of Medical Sciences (AIIMS), Bhubaneswar. **Debasish Basu**– Professor, Postgraduate Institute of Medical Education and Research, Chandigarh, India. **Deepti Dabar**– Additional Professor, All India Institute of Medical Sciences (AIIMS), Bhopal. **Forhad Akhtar Zaman** – Additional Professor, All India Institute of Medical Sciences (AIIMS), Guwahati. **Johnson Pradeep Ruben** – Professor and Head, Department of Psychiatry, St. John's Medical College Hospital, Bengaluru. **Limalemla Jamir** – Associate Professor, All India Institute of Medical Sciences (AIIMS), Guwahati. **Neha Dahiya**– Scientist D, Division of Delivery Research, Indian Council of Medical Research, New Delhi. **Neha Purohit**- Senior Research Officer, Postgraduate Institute of Medical Education and Research, Chandigarh. **Prem Mony** – Professor & Head, Division of Epidemiology, Biostatistics and Population Health. **Priyamadhaba Behera** – Associate Professor, AIIMS Bhubaneswar. **Pulkit Verma** – Scientist D, Data Centre, Indian Council of Medical Research, New Delhi. **Rakesh Kumar**– Associate Professor, All India Institute of Medical Sciences (AIIMS), New Delhi. **Ramdas Ransing** – Associate Professor, All India Institute of Medical Sciences (AIIMS), Guwahati. **Rashmi Agarwalla** – Associate Professor, All India Institute of Medical Sciences (AIIMS), Guwahati. **Renjith R Pillai**– Additional Professor, Postgraduate Institute of Medical Education and Research, Chandigarh. **Rohit Verma** – Professor, All India Institute of Medical Sciences (AIIMS), New Delhi. **Roshan Fakirchand Sutar** – Associate Professor, All India Institute of Medical Sciences (AIIMS), Bhopal. **Sanjay Agarwal** – Professor and Head of Department, Atal Bihari Vajpayee Government Medical College, Vidisha. **Shree Mishra** – Associate Professor, All India Institute of Medical Sciences (AIIMS), Bhubaneswar. **Snehil Gupta**– Associate Professor, All India Institute of Medical Sciences (AIIMS), Bhopal. **Shankar Prinja** – Professor, Postgraduate Institute of Medical Education and Research, Chandigarh, India. **Sharad Philip**– Assistant Professor, All India Institute of Medical Sciences (AIIMS), Guwahati. **Siddharth Sarkar—**Additional Professor, All India Institute of Medical Sciences (AIIMS), New Delhi. **Sofia Mudda** – Medical Officer, All India Institute of Medical Sciences (AIIMS), Bhopal. **Yashdeep Gupta** – Professor, All India Institute of Medical Sciences (AIIMS), New Delhi. **Yatan Pal Singh Balhara**– Professor, All India Institute of Medical Sciences (AIIMS), New Delhi

**Mentor Group: Atreyi Ganguli** – National Professional Officer, Mental Health & Substance Abuse and Healthy Ageing, WHO India. **Krishnamurthy Jayanna** – Director, Implementation Research Expert, Centre for Integrative Health & Wellbeing, Ramaiah University of Applied Science, Bengaluru. **Preeti Sinha** – Professor, Geriatric Psychiatric Unit, Department of Psychiatry, NIMHANS, Bengaluru. **Rajesh Sagar** – Professor, Dept of Psychiatry, AIIMS, New Delhi. **R M Pandey** – A S Paintal Distinguished Scientist Chair, ICMR, NIMS, New Delhi. **Sharmila Mazumdar** – Implementation Research Expert, Society for Applied Studies, Delhi. **Suneela Garg** – Chair, Program Advisory Committee, NIHFW, Delhi. **Vivek Agarwal** – Professor & Head, King George Medical University, Lucknow. **Vivek Benegal** – Professor, Dept of Psychiatry, NIMHANS, Bengaluru.

## Author contributions

**Conceptualization:** Johnson-Pradeep Ruben, Yatan Pal Singh Balhara, Ajay Chauhan, Abhishek Ghosh, Limalemla Jamir, Anindo Majumdar, Arpit Parmar, Debasish Basu, Prem Mony, Chirag Parmar, Pulkit Verma, Neha Dahiya, Ashoo Grover.

**Data curation:** Johnson-Pradeep Ruben, Yatan Pal Singh Balhara, Abhishek Ghosh, Limalemla Jamir, Anindo Majumdar, Arpit Parmar, Debasish Basu, Ajay Chauhan, Chirag Parmar.

**Formal analysis:** Johnson-Pradeep Ruben, Yatan Pal Singh Balhara, Abhishek Ghosh, Limalemla Jamir, Anindo Majumdar, Arpit Parmar, Debasish Basu, Prem Mony, Chirag Parmar, Renjith-R Pillai, Pulkit Verma, Neha Dahiya, Ashoo Grover.

**Funding acquisition:** Abhishek Ghosh, Limalemla Jamir, Anindo Majumdar, Arpit Parmar, Johnson-Pradeep Ruben, Yatan Pal Singh Balhara, Debasish Basu, Ajay Chauhan.

**Investigation:** Johnson-Pradeep Ruben, Yatan Pal Singh Balhara, Ajay Chauhan, Abhishek Ghosh, Limalemla Jamir, Anindo Majumdar, Arpit Parmar, Debasish Basu, Chirag Parmar, Sharad Philip, Renjith-R Pillai, Shankar Prinja, Siddharth Sarkar, Roshan Fakirchand Sutar, Pulkit Verma, Neha Dahiya.

**Methodology:** Johnson-Pradeep Ruben, Yatan Pal Singh Balhara, Ajay Chauhan, Abhishek Ghosh, Limalemla Jamir, Anindo Majumdar, Arpit Parmar, Debasish Basu, Prem Mony, Chirag Parmar, Sharad Philip, Renjith-R Pillai, Shankar Prinja, Siddharth Sarkar, Roshan Fakirchand Sutar, Pulkit Verma, Neha Dahiya, Ashoo Grover.

**Project administration:** Neha Dahiya, Ashoo Grover, Pulkit Verma,  Johnson-Pradeep Ruben, Yatan Pal Singh Balhara, Abhishek Ghosh, Limalemla Jamir, Anindo Majumdar, Arpit Parmar, Debasish Basu, Ajay Chauhan.

**Resources:** Johnson-Pradeep Ruben, Yatan Pal Singh Balhara, Ajay Chauhan, Abhishek Ghosh, Limalemla Jamir, Anindo Majumdar, Arpit Parmar, Debasish Basu, Sharad Philip, Renjith-R Pillai, Shankar Prinja, Siddharth Sarkar, Roshan Fakirchand Sutar, Pulkit Verma, Neha Dahiya, Ashoo Grover.

**Software:** Pulkit Verma, Yatan Pal Singh Balhara, Neha Dahiya.

**Supervision:** Neha Dahiya, Ashoo Grover, Pulkit Verma,  Abhishek Ghosh, Limalemla Jamir, Johnson-Pradeep Ruben, Yatan Pal Singh Balhara, Anindo Majumdar, Ajay Chauhan, Arpit Parmar.

**Writing – original draft:** Johnson-Pradeep Ruben, Yatan Pal Singh Balhara, Neha Dahiya, Shankar Prinja, Neha Purohit, Pulkit Verma.

**Writing – review & editing:** Johnson-Pradeep Ruben, Yatan Pal Singh Balhara, Ajay Chauhan, Abhishek Ghosh, Limalemla Jamir, Anindo Majumdar, Arpit Parmar, Debasish Basu, Debadatta Mohapatra, Chirag Parmar, Sharad Philip, Renjith-R Pillai, Shankar Prinja, Siddharth Sarkar, Roshan Fakirchand Sutar, Pulkit Verma, Neha Dahiya.

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
