## [Decision Letter · Decision Letter 0]

18 Jul 2025

Dear Dr. Dahiya,

Thank you for submitting your manuscript to PLOS ONE. After careful consideration, we feel that it has merit but does not fully meet PLOS ONE’s publication criteria as it currently stands. Therefore, we invite you to submit a revised version of the manuscript that addresses the points raised during the review process.

We look forward to receiving your revised manuscript.

Kind regards,

Hariom Kumar Solanki, M.D.

Academic Editor

PLOS ONE

Journal Requirements:

“Indian Council of Medical Research”

4. One of the noted authors is a group or consortium “ICMR MINDS Study Group”. In addition to naming the author group, please list the individual authors and affiliations within this group in the acknowledgments section of your manuscript. Please also indicate clearly a lead author for this group along with a contact email address.

6. Please upload a copy of Figures 2 and 3, to which you refer in your text on page 17 and 52. If the figure is no longer to be included as part of the submission please remove all reference to it within the text.

7. We note you have included a table to which you do not refer in the text of your manuscript. Please ensure that you refer to Table 11 and 12 in your text; if accepted, production will need this reference to link the reader to the Table.

Reviewers' comments:

Reviewer's Responses to Questions

**Comments to the Author**

1. Does the manuscript provide a valid rationale for the proposed study, with clearly identified and justified research questions?

Reviewer #1: Yes

Reviewer #2: Partly

2. Is the protocol technically sound and planned in a manner that will lead to a meaningful outcome and allow testing the stated hypotheses?

Reviewer #1: Yes

Reviewer #2: Partly

3. Is the methodology feasible and described in sufficient detail to allow the work to be replicable?

Reviewer #1: Yes

Reviewer #2: Yes

4. Have the authors described where all data underlying the findings will be made available when the study is complete?

Reviewer #1: Yes

Reviewer #2: Yes

5. Is the manuscript presented in an intelligible fashion and written in standard English?

Reviewer #1: Yes

Reviewer #2: Yes

You may also provide optional suggestions and comments to authors that they might find helpful in planning their study.

Reviewer #1: attached

Reviewer #2: The title could be further improved.

Page 8: If there are aims and objectives, they are to be separated.

Page 9: The sentence ‘mixed-methods quasiexperimental pre-post (without a control group, within-site, three phases, single-arm, interrupted time series design’ requires improvement in terms of clarity. E.g. A mixed-methods, single-arm, quasi-experimental interrupted time series design with a pre-post approach conducted within-site over three phases and without a control group.

Page 9: around 80 health facilities, the word around to be omitted.

The detailed inclusion and exclusion criteria for the participants are to be specified.

Page 10: Hqrs. The full name/form needs to be provided.

A flow chart of the study design for the study (overall study) is to be provided e.g., multistage sampling etc

The sampling method is to be presented.

The primary outcome(s) and secondary outcome of the study are to be clearly stated. It is unclear if the focus is on mental health and substance use disorder. Although mental health and substance use disorders (SUDs) often co-occur and influence each other, they can occur independently.

Page 11: 70% coverage*, what is the purpose of the asterisk symbol?

Page 11: The sample size calculation is unclear, e.g., whether to have coverage 70% or more than 70% is to be clarified. The aspect of sample size calculation requires improvement and clarity, and consideration of the design effect/ICC parameter if appropriate.

Page 12: To achieve the same, the word ‘the same’ is to be replaced with the word ‘this’

The inventories/questionnaires/tools, whether in English or Tamil version as well as the language communication; validation information, self-administered or interview are to be stated.

Figure 2: No figure included.

Figure 3 Conceptual framework for costing of intervention package: No figure included.

Page 22: M/SUD or MSUD?

Page 36, 38: There are dot symbols without text in the table and need to be omitted.

Page 46 'baseline;': symbol; is to be omitted.

Page 57: The versions of the IBM SPSS software and R software are to be stated. One or two-tailed test is to be stated.

Page 57: The statement ‘paired t-tests or their nonparametric variants, will be applied for testing association,’ is incorrect.

Page 57: The sentence ‘ For comparing proportions among dependent groups, the McNemar test will be used.’ incomplete and can be improved. e.g. dependent (paired) groups.

Page 57: The section 'To compare the mean scores on various outcome measures between baseline and post-intervention independent sample t-test, one-way ANOVA, and one-way and two-way repeated measures ANOVA will be used. Appropriate statistical tests for categorical and quantitative variables, like chi-square, McNemar, and paired t-tests or their nonparametric variants, will be applied for testing association, depending on whether data is normally or non-normally distributed.' lacks clarity and requires revision. For each statistical test mentioned, its purpose and the type of comparison it is used for, is to be clearly stated.

Page 57: The sentence ‘Cluster analysis techniques will be done if clustering of factors is identified during data analysis.’ could be improved e.g. If clustering of variables is detected during analysis, cluster analysis techniques will be employed to explore these patterns.

Page 57: The version for ATLAS.ti software is to be stated. The statement ‘Qualitative data management will be done using ATLAS.ti software.’ could be improved. e.g. ATLAS.ti 25 will be used for qualitative data analysis, as well as data management.

Page 57 and other sections: The cost analyses is unclear. The type of cost analyses is to be stated. Cost analyses and economic evaluation are two separate things. e.g., economic evaluation – CEA, CBA etc.

Supplemental 1, 3 and Table 11 are to be cited in the text.

Some references did not conform to the journal format.

There was no SPIRIT checklist attached.

There are alignment issues with the text in tables and text and typographical issues, e.g., spacing, cap, or small caps, etc that require thorough proofreading.

**Do you want your identity to be public for this peer review?** For information about this choice, including consent withdrawal, please see our Privacy Policy

Reviewer #1: No

Reviewer #2: No

---

## [Author Response · Author response to Decision Letter 1]

25 Aug 2025

Response to Reviewers

Manuscript PONE-D-25-23341 titled “ICMRs Multistate Implementation Research Study on Integration of Screening and Management of Mental and Substance Use Disorders with Other Non-Communicable Diseases (ICMR-MINDS) – An Implementation Research Study Protocol”

We would like to thank the editorial team and the reviewers for their valuable time and efforts for the comments. Kindly find attached the rebuttal for the comments.

Main Heading Reviewer’s Queries Response to Queries Page and Line Nos

Journal Requirements:

1.Please ensure that your manuscript meets PLOS ONE's style requirements, including those for file naming. The PLOS ONE style templates can be found at https://journals.plos.org/plosone/s/file?id=wjVg/PLOSOne_formatting_sample_main_body.pdf and https://journals.plos.org/plosone/s/file?id=ba62/PLOSOne_formatting_sample_title_authors_affiliations.pdf

We have downloaded the same and have used the templates for the revised manuscript.

Full Manuscript

2. Please note that funding information should not appear in any section or other areas of your manuscript. We will only publish funding information present in the Funding Statement section of the online submission form. Please remove any funding-related text from the manuscript. We have removed the funding information from the manuscript.

Full Manuscript

“Indian Council of Medical Research”

Please include this amended Role of Funder statement in your cover letter; we will change the online submission form on your behalf. We would like to provide the statement given below and will include it in the cover letter.

The funders have been involved in the study design, decision to publish and preparation of the manuscript. ND, AG, and PV are staff members of ICMR. The opinions expressed by them in this paper are their own and do not necessarily reflect the policy of ICMR.

We have added the same in cover letter

Cover letter

4. One of the noted authors is a group or consortium “ICMR MINDS Study Group”. In addition to naming the author group, please list the individual authors and affiliations within this group in the acknowledgments section of your manuscript. Please also indicate clearly a lead author for this group along with a contact email address. We have listed the individual authors and affiliations within the ICMR MINDS group in the acknowledgements section of the manuscript.

We would like to clarify that both JPR and YSB contributed equally and share joint lead authorship for this manuscript. Page 64-67, lines 912 to 986.

Page 3, Line 58,59

5. Please include your full ethics statement in the ‘Methods’ section of your manuscript file. In your statement, please include the full name of the IRB or ethics committee who approved or waived your study, as well as whether or not you obtained informed written or verbal consent. If consent was waived for your study, please include this information in your statement as well. We have included a statement mentioning the full name of the Institutional ethics committees that approved our study and about the informed consent as mentioned in the above paragraph. Page 59-61 Lines 799 to 831.

6. Please upload a copy of Figures 2 and 3, to which you refer in your text on page 17 and 52. If the figure is no longer to be included as part of the submission please remove all reference to it within the text. We have revised the previous Figures 2 and 3 to Fig 3 and Fig 4.

Figures 3 and 4 are attached

7. We note you have included a table to which you do not refer in the text of your manuscript. Please ensure that you refer to Table 11 and 12 in your text; if accepted, production will need this reference to link the reader to the Table. We have deleted tables 11 and 12.

8. If the reviewer comments include a recommendation to cite specific previously published works, please review and evaluate these publications to determine whether they are relevant and should be cited. There is no requirement to cite these works unless the editor has indicated otherwise. We have reviewed the reviewers’ comments. We got 2 recommendations to cite specific previously published works from Reviewer #1; one of the references was appropriate to our study and it is added, but the second one was not appropriate; hence, we have not included the second reference. Reference No. 3

Page 67, Line-994 to 997

9. Please review your reference list to ensure that it is complete and correct. If you have cited papers that have been retracted, please include the rationale for doing so in the manuscript text, or remove these references and replace them with relevant current references. Any changes to the reference list should be mentioned in the rebuttal letter that accompanies your revised manuscript. If you need to cite a retracted article, indicate the article’s retracted status in the References list and also include a citation and full reference for the retraction notice. We have reviewed all the references and presented them in the reference style as recommended. None of the references are retracted. Page 67 to 73, Lines 988 to 1127.

Reviewer #1 Add this reference here Page No 5.

Causal associations between HbA1c and multiple diseases unveiled through a Mendelian randomization phenome-wide association study in East Asian populations

We have added the reference as suggested by Reviewer #1.

1. Han L, Xu S, Chen R, Zheng Z, Ding Y, Wu Z, et al. Causal associations between HbA1c and multiple diseases unveiled through a Mendelian randomization phenome-wide association study in East Asian populations. Medicine. 2025 Mar 14;104(11):e41861. doi: 10.1097/MD.0000000000041861 Reference No. 3

Computational Intelligence-Based Classification System for The Diagnosis of Memory Impairment in Psychoactive Substance Users

Reference should be made (Page 8). I searched and this was appropriate. We carefully reviewed the suggested reference (Zhu, 2024, Journal of Cloud Computing) and discussed it with our group. While the paper is valuable in its focus on computational intelligence for diagnosing memory impairment in psychoactive substance users, its scope is primarily centered on artificial intelligence techniques applied to neuroimaging. Our manuscript, in contrast, addresses the integration of mental health into Comprehensive Primary Health Care (CPHC), with emphasis on health system implementation, task-shifting, and community-level strategies for addressing behavioral risk factors and non-communicable diseases.

Since the Zhu (2024) paper does not directly relate to implementation challenges, health systems integration, or primary care strategies, we believe it is not contextually relevant to our discussion.

There are too many tables (Page 61). Please attach some. We have removed some less essential tables from the main text.

Reviewer #2 The title could be further improved. We thank the reviewer for the suggestion. However, after careful consideration and discussion with the group we decided to retain the current title as it clearly conveys the study’s identity (ICMR-MINDS), scope (integration of mental, substance use, and NCD care), and design (implementation research protocol). We believe this ensures clarity and alignment with the objectives of the manuscript.

Reviewer #2

Page 8: If there are aims and objectives, they are to be separated. We have separated the aims and objectives as per the Reviewer #2 suggestions.

Page 8, Lines 175 to 188

Page 9: The sentence ‘mixed-methods quasiexperimental pre-post (without a control group, within-site, three phases, single-arm, interrupted time series design’ requires improvement in terms of clarity. E.g. A mixed-methods, single-arm, quasi-experimental interrupted time series design with a pre-post approach conducted within-site over three phases and without a control group. We have revised the sentence as per Reviewer #2's suggestion.

Page 8-9 , Lines 191 to 193

Page 9: around 80 health facilities, the word around to be omitted. We have omitted the word “around” as per Reviewer #2’s suggestion. Page 9, line 211

The detailed inclusion and exclusion criteria for the participants are to be specified. We have provided the detailed inclusion and exclusion criteria for the participants. Page 10, 225 to 233

Page 10: Hqrs. The full name/form needs to be provided. We have written the full form of the Hqrs.

Page 9, Line 217

A flow chart of the study design for the study (overall study) is to be provided e.g., multistage sampling etc

The sampling method is to be presented. We will be using a multistage sampling method, and the flowchart for the multistage sampling has been provided in the Fig 1 The write-up for the same is provided on pages 12 and lines 267 to 284.

The primary outcome(s) and secondary outcome of the study are to be clearly stated. It is unclear if the focus is on mental health and substance use disorder. Although mental health and substance use disorders (SUDs) often co-occur and influence each other, they can occur independently.

We have added the primary and secondary outcomes clearly. The focus of this study is on both mental health and substance use disorders, and we are looking at it from the perspective of common mental disorders. This is an implementation study, and hence, it has multiple outcomes, which will look at the key process outcomes, implementation outcomes, patient/clinical outcome measures, and service outcomes. These have been mentioned in Tables 6 to 9.

Page 41 Line 566- 577

Page 31 to 48. Tables 6 to 9.

Page 11: 70% coverage*, what is the purpose of the asterisk symbol? The asterisk symbol was inserted by mistake. We apologize for the same. We have removed it Page 11, line 252.

Page 11: The sample size calculation is unclear, e.g., whether to have coverage 70% or more than 70% is to be clarified. The aspect of sample size calculation requires improvement and clarity, and consideration of the design effect/ICC parameter if appropriate.

We thank the reviewer for this point and apologize for the lack of clarity. We clarify the intent and calculations as follows. The evaluation was powered to estimate a facility-level coverage of 70% (i.e., “at least 70%”) with an absolute precision of ±10% and 95% confidence. We used the standard single-proportion formula.

n = Z² p (1−p) / d2

with Z=1.96,p=0.70, and d=0.10, giving n= 80.7, which was rounded up to 81 health facilities per district.

Operationally we aimed for cascade step-performances of ≈90% at each stage (screen → refer → manage). For clarity, 0.9×0.9×0.9=0.729 (≈72.9%), not exactly 70%—i.e., the cascade target is slightly conservative relative to the 70% target used in the sample-size calculation.

With respect to clustering and the design effect, the chosen unit of analysis for the primary endpoint is the health facility (facility-level coverage), so the simple-proportion formula above is appropriate because facilities are sampled as independent units and the outcome is the facility’s observed coverage proportion. If instead the primary analysis were to be carried out at the individual client level (or if clustering within facilities must be explicitly accounted for), sample size should be inflated by a design effect DEFF=1+(m−1) ρ (where m = average number of individuals observed per facility and ρ = ICC). Because ICCs for these facility-level service indicators are generally small but uncertain, we present a sensitivity table below showing adjusted sample sizes for plausible DEFF and non-response assumptions.

We will (a) include this clarification in the methods, (b) report both the facility-level calculation and the sensitivity adjustments in the protocol, and (c) plan a small pilot (or use historical data) to estimate an empirical ICC prior to final site selection, and to re-calculate the adjusted sample if needed. Page 11, Lines 252 to 265

Page 12: To achieve the same, the word ‘the same’ is to be replaced with the word ‘this’ As per the above suggestion, while modifying the sample size calculation paragraph, we have modified this particular line. The revised paragraph is as above

The inventories/questionnaires/tools, whether in English or Tamil version as well as the language communication; validation information, self-administered or interview are to be stated. The inventories/questionnaires/tools are available in English and local languages of each state. Those tools created by us will be validated in the field by doing a pilot study in each site. All the tools will be administered in an interview format. Page 13, lines 304 - 305

Figure 2: No figure included. We have revised Fig 2 to Fig 3 and it is attached. Fig 3

Figure 3 Conceptual framework for costing of intervention package: No figure included. We have revised Fig 3 to Fig 4 and it is attached. Fig 4

Page 22: M/SUD or MSUD? We have replaced all the M/SUD to MSUD. Page 17,18; line 397,398, 406

Page 36, 38: There are dot symbols without text in the table and need to be omitted. We have deleted the same.

Page 46 'baseline;': symbol; is to be omitted. We have deleted the same.

Page 57: The versions of the IBM SPSS software and R software are to be stated. We have added the relevant versions of the software. Page 58, Line 764 -765

One or two-tailed test is to be stated. We will be using a two-tailed test and have mentioned in the text Page 59, line 783

Page 57: The statement ‘paired t-tests or their nonparametric variants, will be applied for testing association,’ is incorrect. We have revised the above statement to make the sentences clearer as per the Reviewer’s suggestions.

“Appropriate statistical analyses will be conducted based on the type of data and study design. Comparisons of mean scores between two independent groups will be performed using the independent samples t-test or the nonparametric equivalent if the assumption of normality is violated. For comparisons involving more than two independent groups, one-way ANOVA will be employed. To assess changes in outcome measures over time within subjects, one-way or two-way repeated measures ANOVA will be used, depending on the number of within- and between-subject factors. For categorical variables, associations between independent groups will be examined using the Chi-square test, while the McNemar test will be applied to assess changes in paired categorical data (e.g., pre- and post-intervention). Comparisons of paired quantitative data, such as baseline and post-intervention measurements within the same group, will be conducted using the paired t-test or its nonparametric alternative, such as the Wilcoxon signed-rank test, depending on data distribution. A two-tailed test will be employed”

Page 58 and 59, lines 770 to 783

Page 57: The sentence ‘ For comparing proportions among dependent groups, the McNemar test will be used.’ incomplete and can be improved. e.g. dependent (paired) groups.

Page 57: The section 'To compare the mean scores on various outcome measures between baseline and post-intervention independent sample t-test, one-way ANOVA, and one-way and two-way repeated measures ANOVA will be used. Appropriate statistical tests for categorical and quantitative variables, like chi-square, McNemar, and paired t-tests or their nonparametric variants, will be applied for testing association, depending on whether data is normally or non-normally distributed.' lacks clarity and requires revision. For each statistical test mentioned, its purpose and the type of comparison it is used for, is to be clearly stated.

Page 57: The sentence ‘Cluster analysis techniques will be done if clustering of factors is identified during data analysis.’ could be improved e.g. If clustering of variables is detected during analysis, cluster analysis techniques will be employed to explore these patter

---

## [Decision Letter · Decision Letter 1]

1 Sep 2025

ICMRs Multistate Implementation Research Study on Integration of Screening and Management of Mental and Substance Use Disorders with Other Non-Communicable Diseases (ICMR-MINDS) – An Implementation Research Study Protocol

PONE-D-25-23341R1

Dear Dr. Dahiya,

We’re pleased to inform you that your manuscript has been judged scientifically suitable for publication and will be formally accepted for publication once it meets all outstanding technical requirements.

Kind regards,

Hariom Kumar Solanki, M.D.

Academic Editor

PLOS ONE

Additional Editor Comments (optional):

Reviewer #2:

Reviewers' comments:

Reviewer's Responses to Questions

**Comments to the Author**

1. Does the manuscript provide a valid rationale for the proposed study, with clearly identified and justified research questions?

Reviewer #2: Yes

2. Is the protocol technically sound and planned in a manner that will lead to a meaningful outcome and allow testing the stated hypotheses?

Reviewer #2: Yes

3. Is the methodology feasible and described in sufficient detail to allow the work to be replicable?

Reviewer #2: Yes

4. Have the authors described where all data underlying the findings will be made available when the study is complete?

Reviewer #2: Yes

5. Is the manuscript presented in an intelligible fashion and written in standard English?

Reviewer #2: Yes

You may also provide optional suggestions and comments to authors that they might find helpful in planning their study.

Reviewer #2: The authors have adequately addressed the previous comments. I have no further concerns, and the manuscript is suitable for publication.

**Do you want your identity to be public for this peer review?** For information about this choice, including consent withdrawal, please see our Privacy Policy

Reviewer #2: No

---

## [Editor Report · Acceptance letter]

PONE-D-25-23341R1

PLOS ONE

Dear Dr. Dahiya,

I'm pleased to inform you that your manuscript has been deemed suitable for publication in PLOS ONE. Congratulations! Your manuscript is now being handed over to our production team.

Kind regards,

on behalf of

Dr. Hariom Kumar Solanki

Academic Editor

PLOS ONE